# Evolutionary Architecture Search Through Grammar-Based Sequence Alignment

## Abstract

Neural architecture search (NAS) in expressive search spaces is a computationally hard problem, but it also holds the potential to automatically discover completely novel and performant architectures. To achieve this we need effective search algorithms that can identify powerful components and reuse them in new candidate architectures. In this paper, we introduce two adapted variants of the Smith-Waterman algorithm for local sequence alignment and use them to compute the edit distance in a grammar-based evolutionary architecture search. These algorithms enable us to efficiently calculate a distance metric for neural architectures and to generate a set of hybrid offspring from two parent models. This facilitates the deployment of crossover-based search heuristics, allows us to perform a thorough analysis on the architectural loss landscape, and track population diversity during search. We highlight how our method vastly improves computational complexity over previous work and enables us to efficiently compute shortest paths between architectures. When instantiating the crossover in evolutionary searches, we achieve competitive results, outperforming competing methods. Future work can build upon this new tool, discovering novel components that can be used more broadly across neural architecture design, and broadening its applications beyond NAS.

## 1 Introduction

Neural architecture search (NAS) (Elsken et al., 2019) has traditionally operated in constrained search spaces, defined by limited operations and fixed topologies. Popular benchmarks such as NAS-Bench-101 (Ying et al., 2019) and NAS-Bench-201 (Dong and Yang, 2020) restrict exploration to cell-based architectures built from only a handful of primitives. While these constraints simplify optimisation, they confine the search to narrow structural templates that cannot lead to fundamentally better architectures but rather incremental improvements of existing ones.

Recently, more expressive NAS search spaces have been proposed to enable broader architectural discovery. For example, Hierarchical NAS (Schrodi et al., 2023) expands cell-based spaces by including high-level macro design choices such as network topology. Similarly, Ericsson et al. (2024) introduce *einspace*, a parameterised context-free grammar (PCFG) that spans architectures of varying depth, branching patterns, and operation types. Unlike earlier spaces that only fine-tuned existing templates, these approaches unlock the potential to discover fundamentally new architectures.

However, this expressiveness comes at the cost of scalability. The vast size and complexity of grammar-based spaces make exploration difficult. Evolutionary operators such as mutation and crossover, while effective in small DAG-based spaces (Real et al., 2019), do not easily transfer to these flexible representations. Moreover, the lack of tractable distance metrics for large graphs or trees hinders the ability to control diversity or measure smoothness within the search. Advancing NAS therefore requires efficient metrics and recombination operators tailored to expressive spaces.

Prior work on shortest edit path crossover (SEPX) demonstrated a theoretically principled method by finding the minimal sequence of graph edits needed to transform one parent into the other (Qiu and Miikkulainen, 2023). This approach addresses the permutation problem where different graph encodings represent equivalent networks. While SEPX can yield high-quality offspring in smaller spaces, it does not scale well to the larger, more complex architectures of modern NAS. Computing the true shortest edit path essentially requires solving a graph edit distance problem, which is NP-hard (Bougleux et al.,

2017). As the number of nodes and connections increases, graph matching becomes computationally intractable (as we show in Figure 3), limiting SEPX's applicability in expressive NAS spaces.

In this work, we propose a scalable alternative: *a novel evolutionary crossover operator inspired by local sequence alignment.* By leveraging the context-free grammar representation introduced in *einspace* (Ericsson et al., 2024), our method represents each architecture as a sequence of tokens and employs a constrained variant of the Smith-Waterman algorithm (Smith and Waterman, 1981) to identify high-scoring local alignments between parent sequences. These alignments serve as a shortest edit path between architectures and can be used to guide recombination, ensuring that offspring inherit coherent and functionally analogous substructures. Furthermore, the edit distance we get is a metric on space of functional architectures and can be used to aid the search itself, controlling diversity, as well as to perform extensive analysis on the architectural loss landscape. Crucially, these benefits can be attained due to the efficient computation that offers orders of magnitude speed-ups compared to SEPX.

- We introduce an efficient grammar-based sequence-alignment algorithm for computing edit paths and distances between neural architectures in expressive search spaces, guaranteeing syntactic validity. We show that this reduces computation time by orders of magnitude compared to previous methods.

- Our algorithm enables powerful new applications: (a) crossover along the shortest edit path in grammar-based NAS, and (b) diversity measurement and architectural loss landscape analysis through its use as a distance metric.

- We demonstrate these applications in one of the most expressive search space to date, establishing new tools for both search and interpretability in NAS. Our analysis reveals the loss landscape at unprecedented scale, quantifying its smoothness and clustered structure.

## 2 BACKGROUND

**Neural architecture search.** NAS is commonly described in terms of three components: a *search space*, which defines the set of candidate architectures; a *search strategy*, which explores that space; and a *performance estimator*, which evaluates candidate models (Elsken et al., 2019). This work focuses on methods that enrich search strategies and enable more effective exploration and analysis of expressive search spaces.

**Grammar-based search spaces.** Context-free grammars (CFGs) provide a compact and expressive way to encode architectures through a set of production rules. The *einspace* framework (Ericsson et al., 2024) builds on this idea by constructing a grammar that allows for complex architecture topologies while keeping a simple set of rules. In particular, the rule

$$\texttt{M} \quad \rightarrow \quad \texttt{B M M A}[1], \tag{1}$$

specifies a `branching(2)` module where two independent submodules (`M`) are combined via a branching operator (`B`) followed by an aggregation operator (`A`). The resulting component allows for branching structures in the network, and thus break the sequential nature of the architectural encodings. In addition to Branching modules, other production rules of the grammar describe Sequential and Routing modules, and terminal nodes can be grouped into types—Branching, Aggregation, Pre-Routing, Post-Routing and Computation. For more details, see Ericsson et al. (2024).

**Evolutionary search.** Evolutionary algorithms search by maintaining a population of candidate architectures and iteratively applying selection and variation operators (Liu et al., 2023). Variation is typically achieved through *mutation*, which introduces small random edits, and *crossover*, which recombines substructures from two parents into new offspring. While mutation drives local exploration, effective crossover can accelerate search by sharing and recombining high-performing architectural motifs across the population.

---

[1]We specifically refer to the branching obtained with two submodules (branching factor of 2), as these are sampled independently from the grammar. For branching factors of 4 or 8, a single submodule is repeated, which does not introduce permutation invariance problems.

## 3 RELATED WORK

Recent NAS methods use context-free grammars (CFGs) to create more expressive architectural search spaces. Hierarchical NAS (Schrodi et al., 2023) uses grammars to compose diverse macro- and micro-structures, expanding significantly beyond traditional cell-based spaces. Ericsson et al. (2024) employs probabilistic CFGs with recursive rules, enabling novel architectures with varied depth, width, and operations, encompassing varied known performant models. Grammar-based NAS significantly enhances architectural expressiveness, enabling diverse structures such as CNNs and Transformer variants in the same space. However, the resulting vast search spaces require specialised strategies—e.g., Bayesian optimisation (Ru et al., 2021) or seeded evolutionary methods (Ericsson et al., 2024)—to effectively navigate them.

The natural encoding for architectures in these spaces is the derivation tree, formed by following the production rules of the grammar to the architecture string sequence. The fastest and simplest way of crossing over such an encoding is subtree crossover (STX), which simply swaps two non-terminal nodes from the parent architectures (Nordin et al., 1998). This facilitates the sharing of well-performing blocks, but its simplicity hinders its ability to discover hybrid blocks. Consequently, the population diversity can stagnate, and both the exploration and exploitation of the architectural space become mainly driven by mutation operators rather than the crossover itself.

Moreover, evolutionary NAS crossover faces the permutation problem, where different representations can encode the same functional architecture.

Traditional methods for ordered tree edit distance, as proposed for example by Zhang and Shasha (1989), fail to address this issue. SEPX (Qiu and Miikkulainen, 2023), on the other hand, addresses the permutation problem by recombining parent architectures via minimal graph edit operations, preserving maximal common structures and ensuring permutation invariance. SEPX outperforms standard evolutionary methods theoretically and empirically (Qiu and Miikkulainen, 2023), but computing exact graph edit distances is NP-hard, restricting its practical use to small-scale graphs (e.g., NAS-Bench-101's 7-node cells). This computational limitation hampers its scalability to larger, more complex architectures. In general, none of the methods proposed in the literature make use of the inherit structures found in the functional neural architectures represented by the corresponding trees and graphs, and thus tend to compute unnecessary alignment options.

An alternative approach to calculate edit paths and distances is to treat the architecture graph as a linearised sequence and apply sequence alignment methods, such as the Needleman-Wunsch (NW) (Needleman and Wunsch, 1970) and Smith-Waterman (SW) algorithms (Smith and Waterman, 1981). This has been studied in the context of NAS by Mateo Avila Pava (2024), who use NW to compute distances between architectures and thus study and control population diversity during search. The main restriction of this approach, however, is that it only works for sequential representations of architectures, often known as chain-structured spaces. This means it cannot be used for more complex network topologies including components like skip connections and multi-head branching. Moreover, the authors do not integrate the edit path into a crossover operator but rather use a simple one-point crossover, which again limits the exploration to sequential architectures.

Our approach achieves the crossover and distance metric abilities of SEPX (Qiu and Miikkulainen, 2023) but at a vastly improved speed due to our treatment of the encoding in a hybrid tree/sequential way. We use fast dynamic programming to deal with the purely sequential part of the architectures, while recursively applying the method on any branching components to tackle permutation invariance. This effectively combines Smith-Waterman with the ordered and constrained unordered tree edit distance algorithms into an efficient method for our setting (Smith and Waterman, 1981; Zhang and Shasha, 1989; Zhang, 1996).

## 4 PROPOSED METHOD

Given a well-defined grammar, any model can be expressed as a sequential set of tokens by serialising the derivation tree. Building on the work by Mateo Avila Pava (2024), we propose two variants of the Smith-Waterman algorithm (Smith and Waterman, 1981) to be used as a computationally efficient edit distance metric and crossover operator. The addition of constraints within the scoring system extends the validity of this method from sequence alignment to ordered trees, yielding our proposed constrained

Smith-Waterman crossover (CSWX) algorithm. CSWX works on a sequential representation of the architectures, presents a high flexibility and computational efficiency, and provides a thorough component-level comparison of the two parent models, alongside the produced offspring.

We further modify CSWX to recursively compute and collapse submatrices corresponding to permuted subsections of the alignment matrix, achieving invariance to permutation, producing the recursive constrained Smith-Waterman crossover (RCSWX) algorithm. This allows the calculation of shortest edit paths and distances between complex graph and semi-ordered tree topologies through grammar-based search space definitions. For further explanation of the subtree crossover (STX) and shortest edit path crossover (SEPX), which will be used to compare the proposed methods with, please refer to Appendix A.

## 4.1 CONSTRAINED SMITH-WATERMAN CROSSOVER

We introduce CSWX in Algorithm 1. Using **Serialise**, we convert each parent model into a simplified sequence. Nodes implied by structure, e.g. *Sequential*, *Grouping*, *Aggregation*, are omitted. To delimit branches and routing nodes, we insert *separator* tokens. To satisfy Smith-Waterman, we prefix each sequence with a *start* token and place the two sequences on perpendicular axes (see Figure 1).

Then, we compute the minimum-cost path that transforms the first parent into the second via additions, deletions, and substitutions. For each operation considered, we check its validity through the **ValidPath** function. This ensures that we only substitute nodes of the same type (e.g., a branching node cannot be replaced by a terminal node). Moreover, if we try to add, delete, or substitute a separator node, it checks that we performed the same operation to the corresponding Routing or Branching node, and that all Routing and Branching nodes we may have added, deleted, or substituted within that path are properly closed by their corresponding separator, avoiding incongruous models. We sequentially fill all positions within the $dist$ and $paths$ matrices, starting from the top left, selecting the path that presents the smallest cost to reach each one. Once the bottom right corner of the matrix is reached, the best path is traced back and transformed into a series of operations through the **TraceBack** function. Following said path from Figure 1 would yield the following offspring.

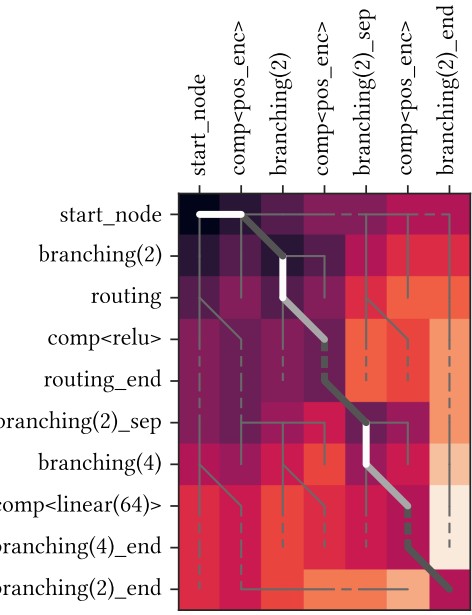

Figure 1: Example $dists$ and $paths$ matrices overlaid. Lighter coloured cells denote a higher distance from the first parent model, shown on top. Moving towards the right entails deleting a node from the first model, moving downwards represents the addition of a node from the second model, and moving diagonally corresponds to a node substitution. Dashed lines signal the closure of branching and routing nodes. The optimal mutation path is traced back in thicker lines, whose brightness reflects the weight of each operation, being brightest intensities a cost of 1, and darker ones reaching a cost of 0.

$$C_{pe},B(2)\{C_{pe},C_{pe}\} \quad \Big\rangle \text{ remove } C_{pe}$$
$$B(2)\{C_{pe},C_{pe}\} \quad \Big\rangle \text{ add } R$$
$$B(2)\{R[C_{pe}],C_{pe}\} \quad \Big\rangle \text{ mutate } C_{pe}{\to}C_{relu}$$
$$B(2)\{R[C_{relu}],C_{pe}\} \quad \Big\rangle \text{ add } B(4)$$
$$B(2)\{R[C_{relu}],B(4)\{C_{pe}\}\} \quad \Big\rangle \text{ mutate } C_{pe}{\to}C_{l64}$$
$$B(2)\{R[C_{relu}],B(4)\{C_{l64}\}\}$$

If we performed all these operations, we would simply obtain the second model; as we are interested in creating a hybrid offspring, we sample a subset of operations through the **SelectOperations** function.

Lastly, these operations are applied within the **GenerateOffspring** function, which produces the desired offspring as the same tree-based format used for the parent models. For visualisations of the architectures that can be constructed along the shortest path in Figure 1, see Appendix B. Further details and pseudocode for all described functions is provided in Appendix C.

---

**Algorithm 1:** Constrained Smith-Waterman Crossover

---

**Input:**    $model1$, the derivation tree for the first architecture to cross over

               $model2$, the derivation tree for the second architecture to cross over

1              $skewness$, the skewness of the operation sampling probability distribution

**Output:** $offspring$, the derivation tree for the resulting hybrid architecture

---

2  $model1_{seq} \leftarrow$ **Serialise**$(model1)$
3  $model2_{seq} \leftarrow$ **Serialise**$(model2)$
4  $dist \leftarrow$ empty array of dimensions length of $model1_{seq} \times$ length of $model2_{seq}$
5  $paths \leftarrow$ empty array of dimensions length of $model1_{seq} \times$ length of $model2_{seq}$
6  **for** $i \leftarrow 1$ **to** length of $model1_{seq}$ **do**
7      **for** $j \leftarrow 1$ **to** length of $model2_{seq}$ **do**
8          $mut, add, rem \leftarrow \infty$
9          **if ValidPath**$(paths, model1_{seq}, model2_{seq}, i, j, "sub")$ **then**
10              $mut \leftarrow dist[i-1,j-1] +$ **SubstitutionCost**$(model1_{seq}[i], model2_{seq}[j])$
11          **if ValidPath**$(paths, model1_{seq}, model2_{seq}, i, j, "add")$ **then**
12              $add \leftarrow dist[i-1,j] + 1 - (model1_{seq}[i]$ is a separator node$)$
13          **if ValidPath**$(paths, model1_{seq}, model2_{seq}, i, j, "rem")$ **then**
14              $rem \leftarrow dist[i,j-1] + 1 - (model2_{seq}[j]$ is a separator node$)$
15          $dist[i,j] \leftarrow \min(mut, add, rem)$
16          $path[i,j] \leftarrow ("sub", "add", "rem")[\mathrm{argmin}(mut,add,rem)]$

17  $ops_{valid} \leftarrow$ **TraceBack**$(model1_{seq}, model2_{seq}, dist, paths)$
18  $ops_{selected} \leftarrow$ **SelectOperations**$(ops_{valid}, skewness)$
19  $offspring \leftarrow$ **GenerateOffspring**$(model1_{seq}, model2_{seq}, ops_{selected})$

---

## 4.2 Recursive Constrained Smith-Waterman Crossover

Collapsing each architecture to a serialised sequence that is fed to CSWX introduces a branch-order permutation problem. As an example, a skip connection around a linear layer $(B(2)\{C_{l64}, C_{id}\})$ is functionally identical regardless of the order of the linear layer and the identity operation in the encoding. The straight-forward approach to handling this is to compute the complete alignment matrix for every combination of branch swaps in either parent architecture. However, this method risks becoming intractable as the number of permutations increases for longer models. We therefore propose a recursive version of CSWX: RCSWX, which reuses all possible pre-computed information in the alignment matrix, and only recalculates the cells that would be affected by the swapping of the branches. RCSWX scales much better on bigger architectures while identifying the same set of operations for generating hybrid offspring as the brute force approach.

RCSWX separates the alignment matrix into submatrices delimited by branching nodes. The number of nested branching nodes for a certain submatrix is referred to as its depth $d$. For each submatrix, we enumerate the $2^d$ possible branch-order permutations as auxiliary submatrices and compute them as in CSWX. At each $separator$ token that closes a branching node, RCSWX collapses the corresponding submatrices into one by retaining, for every cell, the minimum-cost path across all permutations. Note that the collapse does not imply selecting a singular submatrix, but rather combining them by selecting the best path to reach each individual cell in the alignment matrix. Collapsing at the $separator$ tokens enforces the constraints and preserves global optimality, yielding permutation invariance while keeping computation feasible.

### 4.3 Properties of CSWX and RCSWX

#### 4.3.1 Asymptotic Scaling

The SEPX algorithm (Qiu and Miikkulainen, 2023) requires the computation of the Graph Edit Distance (GED) between the two parent architectures, which finds the node correspondence that minimises the total edit cost. The search is combinatorial, thus the number of possible mappings

between two graphs of size $n_1$ and $n_2$ is $n_1^{n_2}$. This $O(n_1^{n_2})$ scaling makes this approach practically intractable for crossover of longer architectures.

In contrast, by treating the architectures as serialised trees instead of graphs we can use dynamic programming through the Smith–Waterman algorithm, which gives us CSWX that scales with

$$T_{CSWX} = O(n_1 n_2). \tag{2}$$

Making CSWX permutation invariant by enumerating all permutations of $b$ branching nodes introduces an exponential factor $2^b$, yielding a total scaling of $O(n_1 n_2 2^b)$ for this brute-force approach. Considering that each branching block is composed of 5 tokens (opening, first branch, separator, second branch, closing), then in the worst case every branch can itself be a branching block. This yields a maximum of $b = (n_1 + n_2)/4$ branch nodes, giving worst-case scaling

$$T_{BF-CSWX} = O\left(n_1 n_2 2^{\frac{n_1+n_2}{4}}\right). \tag{3}$$

RCSWX reduces this exponential factor by reusing partial results within each alignment cell. Instead of a global cost of $2^p$ (with $p$ the maximum nesting depth), the cost locally at a cell $(i,j)$ is only $2^{d_{ij}}$, where $d_{ij}$ is the number of simultaneously open branches. The total runtime is therefore $\sum_{i=1}^{n_1} \sum_{j=1}^{n_2} 2^{d_{ij}}$. The worst case scenario, where both parent models are composed of maximally nested branching blocks along a single branch—which is similar to what may happen in U-Net style networks (Ronneberger et al., 2015)—yields a scaling of

$$T_{RCSWX} = O\left(m 2^{2m}\right) \tag{4}$$

for $m = \frac{n_1}{4} - 1$ assuming $n_1 = n_2$. RCSWX is faster than SEPX and brute-force CSWX in all cases.

### 4.3.2 CSWX AND RCSWX AS DISTANCE METRICS

The edit path computed by (R)CSWX naturally yields an edit distance between two architectures. We consider this distance on two related domains. Let $\mathcal{A}$ denote the set of neural architectures, where two architectures are considered identical if they are functionally equivalent (e.g., differing only by functionally inconsequential permutations of operations). Let $\mathcal{B}$ denote the set of syntactic representations of these architectures in the *einspace* grammar (sequences or trees, where branch order is explicit). We now show that the edit distances defined by CSWX and RCSWX satisfy the axioms of a metric on their respective domains.

**Non-negativity.** Each edit operation has a cost $c_{i,j} \in [0,1]$, so summing over them yields nonnegative edit distances.

**Identity of Indiscernibles.** If two input architectures are identical, the shortest edit path follows the diagonal of the alignment matrix with all mutation costs $c_{i,j} = 0$, giving $d_{\text{CSWX}}(x,x) = 0$ for any $x \in \mathcal{B}$. For non-identical syntactic representations, CSWX may assign a positive cost even when the architectures are functionally equivalent (e.g., differing only by branch permutations). RCSWX resolves this by mapping syntactic forms in $\mathcal{B}$ to their functional counterparts in $\mathcal{A}$, ensuring that

$$d_{\text{RCSWX}}(x,y) = 0 \iff f_{\mathcal{B}\to\mathcal{A}}(x) = f_{\mathcal{B}\to\mathcal{A}}(y), \quad x,y \in \mathcal{B}. \tag{5}$$

**Symmetry.** Swapping the two input architectures simply transposes the alignment matrix. All node additions will become node deletions and vice-versa, and the final distance will be identical.

**Triangle Inequality.** The edit distance between two architectures is defined as the cost of the optimal alignment path between them. Consider three architectures $x, y, z$. The path from $x$ to $z$ can be decomposed by first aligning $x$ to $y$, and then $y$ to $z$. Concatenating these two valid edit paths yields a feasible (though not necessarily optimal) path from $x$ to $z$ with cost $d(x,y) + d(y,z)$. Since Smith–Waterman always finds the minimal-cost valid alignment under the grammar constraints, the optimal cost from $x$ to $z$ cannot exceed this concatenated cost. Therefore,

$$d(x,z) \le d(x,y) + d(y,z). \tag{6}$$

By satisfying all axioms, $d_{\text{CSWX}} : \mathcal{B} \times \mathcal{B} \to \mathbb{R}_{\ge 0}$ is a metric in the *syntactic* space, while $d_{\text{RCSWX}} : \mathcal{A} \times \mathcal{A} \to \mathbb{R}_{\ge 0}$ is a metric in the *semantic* space of architectures.

# 5 EXPERIMENTS

This section lays out the experiments performed to check the capabilities of the (R)CSWX algorithm, both in terms of exploration capabilities and computational expense. We use a subset of datasets from the Unseen NAS benchmark (Geada et al., 2024). Population sizes, running times and other similar hyperparameters were based off of previous experience on the datasets and explored search spaces (Geada et al., 2024; Ericsson et al., 2024). Experiments at scale ran on JUWELS (Kesselheim et al., 2021) and code is made available online at `https://redacted/for/review` under the MIT license.

## 5.1 SEARCH PERFORMANCE ANALYSIS

We compare the results yielded by (R)CSWX with those obtained using Subtree Crossover (STX) and with no crossover across five different random seeds. We set both the crossover and mutation probabilities to 1.0, while the no crossover method uses only mutation. All (R)CSWX hyperparameters such as substitution weights and operation sampling skewness are left as default—see Appendix C for details. We start with a randomly sampled initial population of 100 architectures, and run the search for 900 more iterations to a total of 1000 architecture evaluations. Each update works in a steady-state fashion—that is, we generate a new offspring model and remove the oldest one. The parent models for offspring generation are chosen by tournament selection.

Figure 2 demonstrates the attained validation performances throughout the search and Table 1 shows the final test scores. They highlight the importance of choosing an appropriate search strategy to explore each individual space: some datasets make good use of the information sharing enabled by crossover—e.g., using CSWX on AddNIST or STX on Chesseract—while others benefit from a less constrained exploration—e.g., mutation-only searches on Isabella. In some cases, relying on RCSWX's permutation invariant interpolation underperformed compared to the ones using CSWX, which inject some noise into the shortest path in the form of unnecessary mutation operations. All methods achieved similar validation behaviour on average, but mutation-based approaches show higher overfitting as their test scores are lowest.

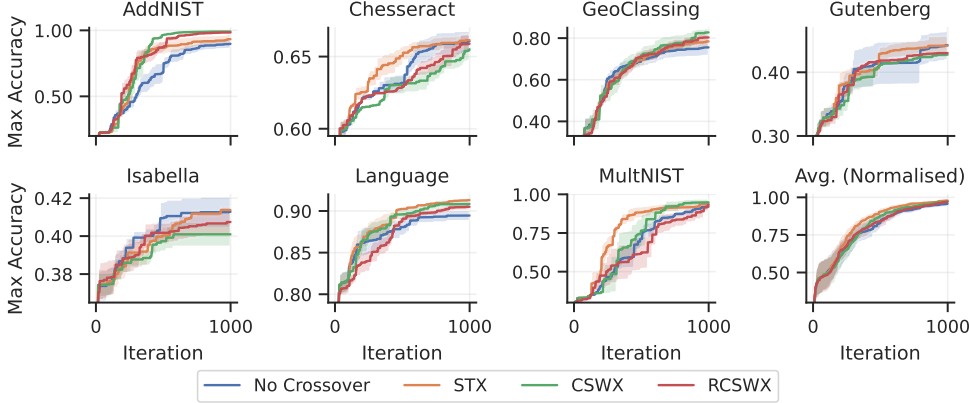

Figure 2: Search results comparing STX, CSWX and RCSWX-based evolutionary search with mutation-only searches. The plots show the mean across five seeds and the standard error of the mean. To assess the average performance, we normalise the results based on the lowest and highest attained performance within each dataset and combine them into a single plot by averaging (bottom right).

Table 1: Test performances of the best models found on the validation set during search, reporting the mean and standard deviation across five seeds. Significance testing can be found in Appendix G.

| Dataset | AddNIST | Chesseract | GeoClassing | Gutenberg | Isabella | Language | MultNIST | Avg. |
|---|---|---|---|---|---|---|---|---|
| No Crossover | 82.13 ± 11.46 | 60.10 ± 0.78 | 79.05 ± 2.44 | 43.35 ± 1.49 | 48.79 ± 2.25 | 93.59 ± 1.06 | 87.99 ± 3.29 | 70.71 ± 1.79 |
| STX | 97.07 ± 0.30 | 60.91 ± 0.49 | 86.27 ± 2.03 | 42.77 ± 1.07 | 49.94 ± 3.81 | 96.16 ± 0.61 | 91.68 ± 2.41 | 74.97 ± 0.73 |
| CSWX | 90.64 ± 4.12 | 58.80 ± 0.82 | 82.80 ± 3.29 | 45.75 ± 1.37 | 53.25 ± 1.83 | 95.95 ± 0.41 | 88.94 ± 2.89 | 73.73 ± 0.93 |
| RCSWX | 95.82 ± 0.36 | 59.94 ± 0.52 | 85.90 ± 2.58 | 44.84 ± 0.41 | 47.27 ± 3.48 | 95.41 ± 0.55 | 91.87 ± 1.22 | 74.44 ± 0.66 |

## 5.2 SCALABILITY OF CONSTRAINED SMITH-WATERMAN CROSSOVER

To prove the computational tractability of the proposed (R)CSWX, the architectures generated in Experiment 5.1 were sorted according to their number of nodes. Then, crossovers were generated using SEPX, CSWX and RCSWX algorithms, whose runtimes are shown in Figure 3. Note that SEPX and RCSWX are guaranteed to produce the same edit paths as they are applied to the same graphs—that is, the *einspace* decision trees—given that they employ the same scoring matrix for the addition, deletion and mutation of layers. We have confirmed this empirically, as all edit paths resulting from SEPX in Figure 3 are identical to those calculated using RCSWX. Maintaining the same operation sampling strategy would also yield the same offspring and, thus, equivalent search results when applied to NAS.

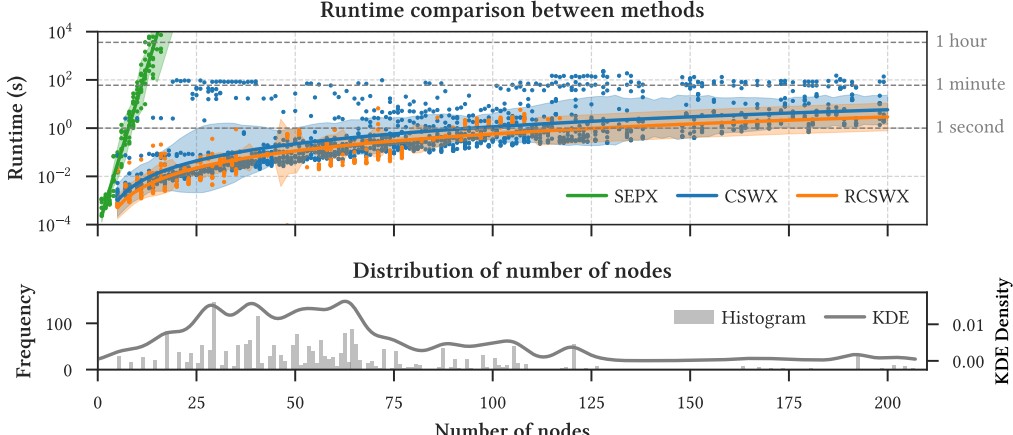

Figure 3: Runtime (s, log scale) against node count for RCSWX, CSWX and SEPX. Scatter markers show raw measurements. Solid lines are global fits: power-law for (R)CSWX and exponential for SEPX, with shaded regions showing adaptive error bounds from local log-residual standard deviations. Extrapolated fits suggest increasing runtime with model complexity. Bottom plot shows the realistic distribution of node counts we see in our searches, highlighting the unfeasibility of SEPX in this setting.

We can see that SEPX quickly becomes intractable at around 20 nodes, while CSWX and RCSWX can be efficiently computed in less than a second for many input architecture pairs. RCSWX even gets to 71 nodes before any computation takes longer than one second. We also see that while the asymptotic scaling of RCSWX as shown above is exponential in the number of nested branching structures, this is not a bottleneck in practice.

## 5.3 SMOOTHNESS OF THE SEARCH SPACE

As shown above, RCSWX acts as an efficiently computable metric on the search space of architectures. To highlight the benefits of this we use it to analyse the structure, and in particular smoothness, of the architectural loss landscape. We calculate the distance between every pair of the 1000 architectures generated for a single seeded search run on the CIFAR10 dataset, which is a relatively simple and well studied dataset, and on the Isabella dataset, that contains fewer, more complex data samples. The $1000 \times 1000$ matrices of distances, along with the precomputed performances of said models, have been used to generate the following plots.

Figures 4a and 4d show two-dimensional UMAP projections based on the edit-distance matrix. The projections reveal that the explored search spaces are highly fragmented: architectures cluster into a small number of well-separated islands. Within each cluster, models are close in edit distance and tend to exhibit somewhat similar performance, but there is less evidence of continuity across clusters. This suggests that while local neighbourhoods may be smooth, the global search space is disconnected. Some noise can be observed within clusters in both UMAPs, which may correlate to destructive mutations—e.g., disruption of functional blocks, excessive feature condensation, etc—that yield low performing models even within promising regions of the architecture spaces.

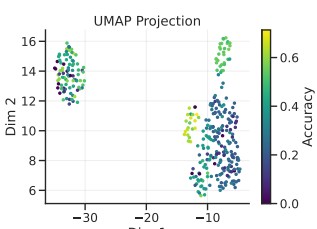 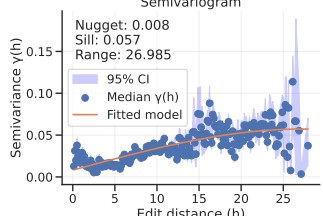 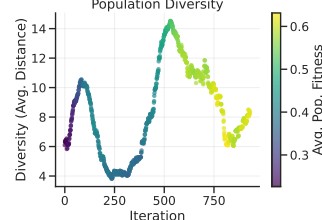

(a) UMAP projection of architectures generated for the CIFAR10 dataset showing clustered structure.

(b) Semivariogram of the population generated for the CIFAR10 dataset with spherical fit.

(c) Diversity of the population generated for the CIFAR10 dataset during the search.

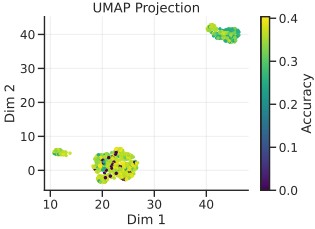 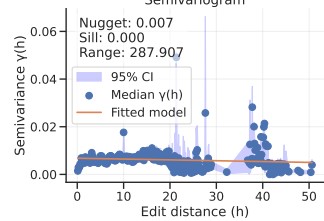 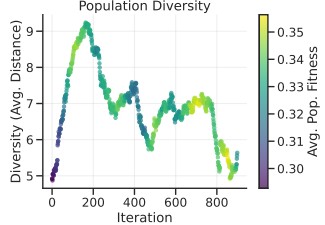

(d) UMAP projection of architectures generated for the Isabella dataset showing clustered structure.

(e) Semivariogram of the population generated for the Isabella dataset with spherical fit.

(f) Diversity of the population generated for the Isabella dataset during the search.

Figure 4: Smoothness and diversity analysis of the architectural search space and search population. (a) and (d) UMAP projections show fragmented global structure but local continuity with some very low performing architectures sprinkled across the biggest clusters. (b) Semivariogram indicates that smoothness in the CIFAR10 space extends to a range of $\sim 25$ edits, beyond which fitness correlation is no longer observable, while (e) semivariogram shows that the distance between two architectures in the Isabella space does not seem to affect how their fitness correlate. (c) Scatter plots of the diversity of the population across the search iterations, measured by the average distance between all pairs of architectures.

To quantify smoothness more formally, we compute semivariograms in Figures 4b and 4e, which measure how performance similarity decays with increasing edit distance. A semivariogram characterises this relationship through three parameters: the *nugget*, capturing variation at zero distance (often due to noise); the *sill*, corresponding to the maximal variance reached; and the *range*, the distance beyond which no correlation remains. In the case of architectures generated to deal with the CIFAR10 dataset, for small edit distances ($h \leq 5$), the semivariance $\gamma(h)$ is very low, indicating that close neighbours differ little in performance. As the distance increases, semivariance rises steadily before starting to plateau around $h \approx 15$. Fitting a spherical model yields a small nugget (consistent with low evaluation noise), a sill matching the maximal performance variance, and a range of roughly 25 edits. This implies that performance correlation is maintained within a neighbourhood of this size, but disappears beyond it. Regarding the Isabella dataset, the correlation appears to be maintained throughout the generated population, indicating that very distinct architectures are expected to attain similar accuracies.

Finally, we use the RCSWX distance metric to measure the continual change in diversity of the population as the search progresses. Figures 4c and 4f indicates that there are significant fluctuations in diversity, with alternating phases of exploration—broadening the search and increasing diversity—and exploitation—converging the population on promising regions of the space. Being able to measure and control this dynamic diversity can be a powerful tool for future search algorithms in these large spaces.

## 6 DISCUSSION AND CONCLUSION

We introduced an efficient algorithm for computing distances and performing crossover in grammar-based NAS search spaces. Our approach is based on a constrained version of the Smith-Waterman algorithm and enables these computations in vastly larger and more complex spaces than previously possible. Our method can be applied to any search space that can be represented as a sequential set

of tokens, and shows very promising results, both in terms of exploration/exploitation capabilities (cf. Figure 2) and computational expense (cf. Figure 3).

Regarding empirical compute times, Figure 3 shows that CSWX can be slower than RCSWX. This may partly stem from implementation overhead, but also from the fact that RCSWX enforces constraints that reduce the number of valid paths, yielding more consistent runtimes. In contrast, CSWX often tracks many longer alignment paths, leading to higher variance and worse scaling. Nonetheless, both methods remain well within acceptable compute times even for large architectures.

CSWX enables crossover-based optimisers to be applied to grammar-based encodings such as *einspace* (Ericsson et al., 2024), while RCSWX adds permutation invariance, making it a valuable distance metric between different architectures and components and, thus, enable the future possibility of controlling diversity during search. Prior work argues that small architectural changes rarely lead to large performance shifts (Yang et al., 2019; Wan et al., 2021), implying that the simpler, less computationally expensive CSWX might be enough to produce near-optimal results. Interestingly, the noise introduced by imperfect model alignment can even be beneficial during architecture searches. Figure 2 shows CSWX sometimes outperforming RCSWX, which may be explained precisely by a higher exploration of the space induced by the absence of permutation invariance, forcing the addition and removal of whole branches during crossover that would otherwise simply be swapped and mutated. With a larger amount of additions and deletions, CSWX may select a very unbalanced set of operations that changes the size (and thus complexity) of the models considerably. This, in turn, broadens the search and is especially advantageous when the initial population is weak or when the search space contains difficult local minima. Imperfect alignment may also result in the addition and deletion of whole functional blocks—mimicking the behaviour of STX—acting like structured mutations that successfully guide exploration given a diverse and performant enough initial population.

RCSWX, by contrast, encourages steadier exploitation by keeping the search focused near the best individuals in the initial population, which may hinder the exploration of novel regions of the architectural space. However, although our initial assessment shows varying performance of the RCSWX-driven genetic algorithm, this crossover tool could be employed to construct better search strategies. For instance, the reuse of functional blocks, which seems to be a very performant strategy, could be mimicked and even improved by (R)CSWX: the alignment matrices provide element-wise addition, deletion and substitution costs, which enable the identification of highly performing sequences that can be preserved during offspring generation or even added to the grammar for subsequent mutation steps. Further research is needed to study how these crossover mechanisms interact with more sophisticated optimisation strategies—such as those described in Appendix F—especially given the exploration–exploitation challenges and the differing behaviour across the loss landscapes examined here. Regardless, our focus in this work is to introduce a theoretically motivated, computationally efficient crossover operator for grammar-based NAS, intended as a tool for further research, rather than as a benchmark for state-of-the-art performance for any given search strategy or exploration-exploitation balance. Although spanning around 140 000 architecture evaluations, our results from seven datasets and five seeds provide only an initial assessment of (R)CSWX-driven searches, showing their potential.

When used as a distance metric, RCSWX allows us to analyse the smoothness of the search space in an unprecedented way. The plots in Figure 4 support the claim that, while sharing common characteristics, architectural search spaces are indeed quite unique. The two datasets analysed in this work, CIFAR10 and Isabella, are both locally smooth, in the sense that small mutations tend to produce architectures of similar quality. However, CIFAR10 is much more globally rugged and fragmented than Isabella, with clusters separated by large distances and little performance correlation across them, while the Isabella search space appears to be much flatter, with little performance difference among clusters. This has direct implications for search: local methods such as crossover-based evolution or hill-climbing can effectively exploit neighbourhoods, but escaping to distant high-performing clusters may require restarts or more exploratory strategies. Indeed, the pure mutation-based strategy yields the fastest convergence on the Isabella loss landscape, presumably because the crossover-based approaches tend to exploit the flat space instead of freely exploring outwards.

Future work can focus on explicitly controlling the exploration-exploitation trade-off by means of the introduced distance metric, implementing and analysing methods like the ones described in Appendix F. We also believe that the shortest paths generated from the crossover method can be used to assign component-level merit within architectures, aiming towards the discovery and reuse of performant architectural blocks and opening up a whole new dimension to explore in the grammar-based NAS field.

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

## A  ALTERNATIVE CROSSOVER METHODS

In this section, the two methods used to compare against throughout the paper are explained in detail. Both methods succesfully generate hybrid offspring, although they differ significantly in complexity and scope.

### A.1  SUBTREE CROSSOVER

Subtree crossover is a classical genetic programming operator that exchanges entire subtrees between parent derivation trees. Koza (1994) demonstrated that this approach preserves coherent functional units, which are essential for maintaining effective computational building blocks. Transferring whole subtrees encourages the reuse of well-performing substructures and supports modular, hierarchical problem solving. Furthermore, its ability to handle variable-length representations allows for a more scalable exploration of complex search spaces. In grammar-based NAS, it is crucial that the exchange subtrees produce syntactically valid architectures. This is achieved by restricting the crossover operation to subtrees rooted at identical non-terminal symbols.

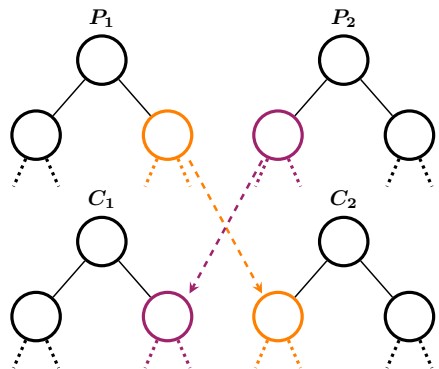

Figure 5: Illustration of subtree crossover. Parent architectures $P_1$ and $P_2$ swap selected subtrees (highlighted), generating two offspring architectures $C_1$ and $C_2$.

Figure 5 on the left provides a visual illustration of subtree crossover. Two parent architectures, $P_1$ and $P_2$, exchange selected subtrees to yield offspring $C_1$ and $C_2$. This procedure swaps portions of the derivation trees while maintaining grammatical validity. Algorithm 2 formalises the process. It begins by extracting node types from each parent and identifying common non-terminal symbols as potential crossover points. If no common non-terminal exists, the crossover is not performed. Otherwise, a random common node is selected from each parent, and the corresponding subtree from one parent replaces that of the other. This approach preserves key structural components and ensures that the resulting architecture adheres to the prescribed grammar.

---

**Algorithm 2:** Subtree Crossover

**Input:**  $model1$, the derivation tree for the first architecture to cross over

$model2$, the derivation tree for the second architecture to cross over

**Output:**  $offspring$, the derivation tree for the resulting hybrid architecture

1  $model1_{types} \leftarrow$ the type of each node in $(model1)$
2  $model2_{types} \leftarrow$ the type of each node in $(model2)$
3  $common_{types} \leftarrow \{node_t \mid node_t \in model1_{seq} \cap model2_{seq}, node_t \text{ is a non-terminal symbol}\}$
4  **if** $common_{types} = \emptyset$ **then**
5  ⌊ **return failure** (no common non-terminals)
6  Randomly select $node_t \in common_{types}$
7  $pos_1 \leftarrow$ the position of a random occurrence of $node_t$ in $model1_{types}$
8  $pos_2 \leftarrow$ the position of a random occurrence of $node_t$ in $model2_{types}$
9  $node_1 \leftarrow$ **ExtractSubtree**$(model1, pos_1)$
10  $node_2 \leftarrow$ **ExtractSubtree**$(model2, pos_2)$
11  $offspring \leftarrow$ **ReplaceSubtree**$(model1, pos_1, node_2)$
12  **return** $offspring$

---

## A.2 SHORTEST EDIT PATH CROSSOVER

Shortest edit path crossover (SEPX) (Qiu and Miikkulainen, 2023) is designed to overcome the permutation problem in evolutionary NAS. Both parent architectures, $model1$ and $model2$, are represented as graphs, where a graph edit operation modifies the graph by inserting, deleting, or substituting a node or an edge. The graph edit distance (GED) is defined as the minimum total cost (with unit cost per operation) required to transform one graph into an isomorphic copy of the other, so that GED equals the length of the shortest edit path between the two graphs.

Formally, let

$$ops^* = \operatorname*{argmin}_{ops_i \in ops} \sum_{j=1}^{|ops_i|} cost(ops_{i,j}),$$

be the shortest edit path from $model1$ to $model2$, which contains $d^*$ unique edit operations. SEPX generates an offspring graph $offspring$ by applying half of these edits to $model1$, sampled at random out of $ops^*$. That is,

$$offspring = ops^*_{\pi_r(\lceil d^*/2 \rceil)} \circ ops^*_{\pi_r(\lceil d^*/2 \rceil - 1)} \circ \cdots \circ ops^*_{\pi_r(1)}(model1),$$

Here, $\pi_r$ denotes a random permutation of the edit operation indices, and the composition $\circ$ indicates sequential application of the edit operations.

SEPX first computes the GED between the two parent graphs, determining the minimal sequence of edit operations needed to convert one parent into the other. About half of these operations are then randomly chosen and applied to one parent, producing an offspring. By aligning the parent graphs before recombination, the method overcomes the permutation problem-where different genotypes encode the same phenotypes-and preserves shared structural components.

---

**Algorithm 3:** SEPX

**Input:**     $model1$, the graph representation of the first architecture to cross over
              $model2$, the graph representation of the second architecture to cross over
**Output:** $offspring$, the graph representation of the resulting hybrid architecture

1 $ops^* \leftarrow$ **ComputeShortestEditPath**($model1$,$model2$);
2 $ops_{selected} \leftarrow$ a random subset of half of the mutations in $ops^*$
3 $offspring \leftarrow$ **GenerateOffspring**($model1_{seq}, model2_{seq}, ops_{selected}$)
4 **return** $offspring$

---

In summary, the key steps are to **ComputeShortestEditPath** to determine the minimal edit sequence, then perform operation selection by randomly sampling half of these operations, and finally **GenerateOffspring** by applying the selected edits. Standard crossover is often disrupted by the permutation problem because different orderings of identical substructures can lead to destructive recombination. By aligning parent graphs using GED, SEPX preserves coherent functional units, making it both a theoretically principled and practically effective method for combining neural architectures.

## A.3 COMPARISON TO OTHER DISTANCE/(DIS)SIMILARITY AND CROSSOVER OPERATORS

Figure 2 compares (R)CSWX to various other methods in the literature that have been applied in the context of NAS.

Table 2: Comparison with other methods for computing distances between architectures and using them as crossover operators.

| Method | Distance Metric | Crossover Operator | Encoding | Spaces | Scaling |
|---|---|---|---|---|---|
| WL kernel (Weisfeiler–Lehman) | ✗ | ✗ | Graph | Any | $O(h(n_1 + m_1 + n_2 + m_2))$ |
| NWNAS (Needleman–Wunsch) | ✓ | ✗ | Sequence | Chain-based | $O(n_1 n_2)$ |
| GED/SEPX (Graph Edit Distance) | ✓ | ✓ | Graph | Cell-based / general DAGs | $O(n_1^{n_2})$ |
| CSWX | ✓ | ✓ | Ordered Tree | Grammar-based | $O(n_1 n_2)$ |
| RCSWX | ✓ | ✓ | Semi-ordered Tree | Grammar-based | $O\left(\sum_{i=1}^{n_1} \sum_{j=1}^{n_2} 2^{d_{ij}}\right)$ |

# B  EXAMPLE INTERMEDIATE OFFSPRING

In this section, we show the two original models and four offspring architectures—as represented by their derivation trees—that could be generated from the CSWX example in Figure 1.

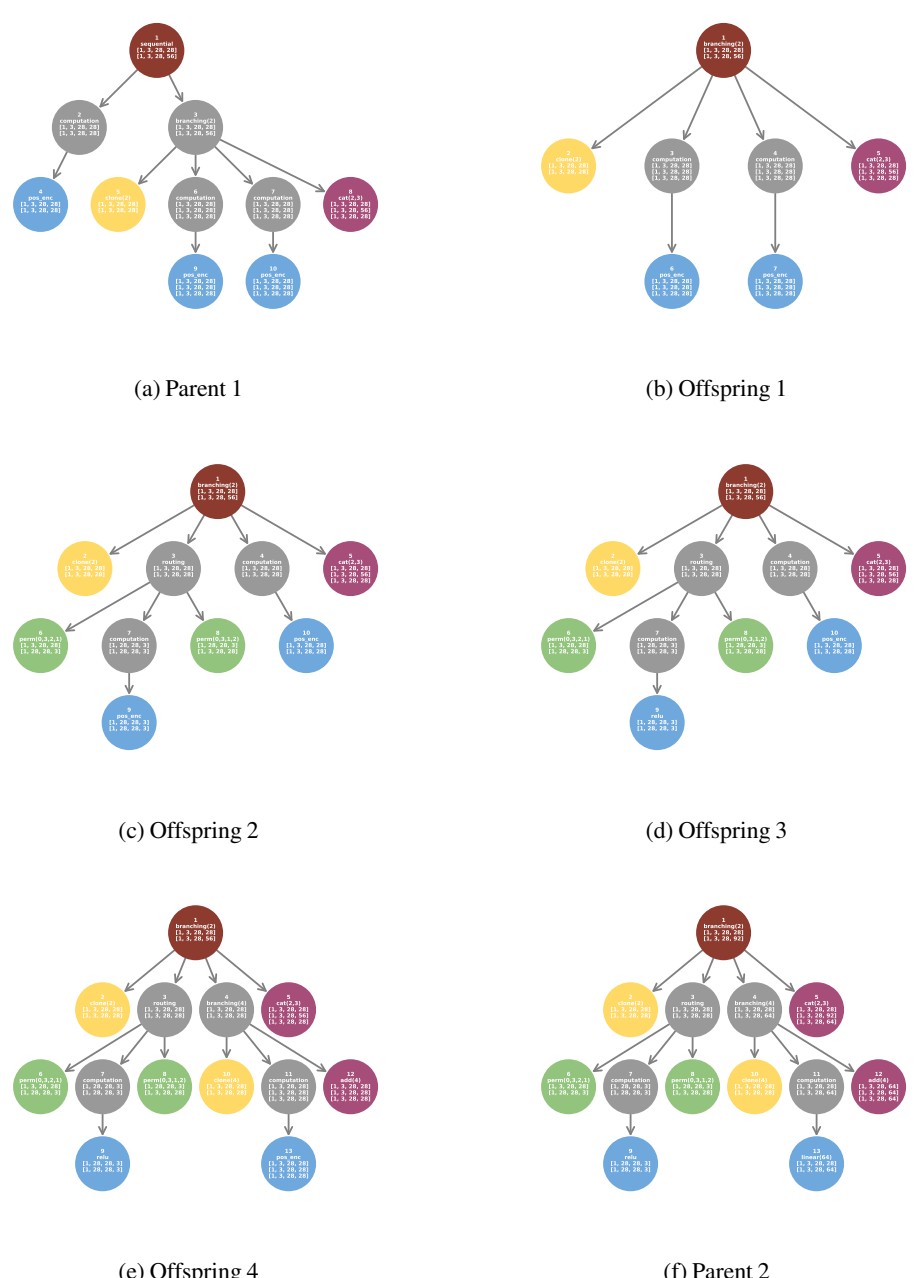

(a) Parent 1

(b) Offspring 1

(c) Offspring 2

(d) Offspring 3

(e) Offspring 4

(f) Parent 2

Figure 6: From left to right, top to down: original model 1 (a), and offspring models resulting from the removal of the leftmost Computation node (b), the addition of a Routing node (c), the mutation of a positional encoding into a ReLU terminal node (d), the addition of a Branching(4) node (e) and mutation of a positional encoding into a linear terminal node (f), resulting in original model 2.

## C  Constrained Smith-Waterman Crossover functions

In this section, a more detailed description of the functions referenced in 1 is laid out.

The **Serialise** function ensures that the resulting list of nodes required to perform CSWX contains the minimal information required while ensuring that the rules of the grammar are preserved. Commencing with a $node_{start}$, it will recursively add the nodes to a list, ignoring Sequential and terminal nodes that can be inferred from the context, and add a $node_{sep}$ after every branch and routing module to signal where it ends. These separator nodes will contain information about the node they are closing and about whether it signals the separation between two branches or the end of the last branch.

There is almost a one-to-one conversion between the tree and the list representation of the models, save for the order in which the Sequential nodes are nested (which makes no difference in the actual architecture of the represented models).

---

**Algorithm 4:** Serialise

**Input:**   $node$, the model or node we want to serialise
**Output:** $node_{seq}$, a serialised representation of $node$

1 **if** $node$ is the root node **then**
2     $node_{seq} \leftarrow node_{start}$
3 **else**
4     $node_{seq} \leftarrow \emptyset$
5 **if** $node$ is not a terminal node **then**
6     **if** $node$ is not a sequential node **then**
7        $node_{seq} \leftarrow node_{seq} \cup node$
8     **for** $child \in$ children of $node$ **do**
9        $node_{seq} \leftarrow node_{seq} \cup$ **Serialise**($child$)
10        **if** $child$ is not a terminal node **and** $node$ is a branching or routing node **then**
11           $node_{seq} \leftarrow node_{seq} \cup node_{sep}$

---

The **SubstitutionCost** function is really straightforward, comparing two nodes to assign a cost of substituting one into the other.

---

**Algorithm 5:** SubstitutionCost

**Input:**   $node1$, the first node to compare
          $node2$, the second node to compare
**Output:** $cost$, the dissimilarity between $node1$ and $node2$

1 **if** $node1$ and $node2$ are the same type of node **then**
2     **if** The first and last children of $node1$ and $node2$ are the same type of node **then**
3        **if** The first and last children of $node1$ and $node2$ have the same hyperparameters **then**
4           $cost \leftarrow 0$
5        **else**
6           $cost \leftarrow 0.25$
7     **else**
8        $cost \leftarrow 0.5$
9 **else**
10     $cost \leftarrow \inf$

---

If the nodes to compare are Computation nodes, the first and last child will be the same—the type of Computation operation—, while for Branching and Routing nodes the first and last children will define the kind of operations performed. For instance, the $node1$ Branching(4)group(1,4), M, cat(1,4) would have:

- **SubstitutionCost**($node1$, Branching(4)group(1,4), M, cat(1,4)) $= 0$

- **SubstitutionCost**($node1$, Branching(8)group(1,8), M, cat(1,8)) $= 0.25$
- **SubstitutionCost**($node1$, Branching(4)clone(4), M, add(4)) $= 0.5$
- **SubstitutionCost**($node1$, Computation(identity) $= \inf$

As of now, the substitution costs are arbitrarily fixed, but in the future they could be modulated by factors such as the probabilities of sampling each type of node from the grammar, the observed impact of substituting one layer into another throughout the search or any given a-priori information, among others. One way of making the mutation cost of performing a mutation that swaps a derivation tree node $y$ for another node $x$ depend solely on the sampling probabilities would be defining it as $C(x,y) = \sum_{i=1}^{N} \left( \prod_{j=1}^{M_i} \left( 1 - P\left(x_i^j\right) \right) \right) \left( 1 - \mathbb{1}_{x_i^j}\left(y_i^j\right) \right)$, where $N$ is the maximum depth of the sampling of the hyperparameters of a node (for instance, Computation $\rightarrow$ linear $\rightarrow$ 64 would have $N = 2$), $M_i$ is the number of options for the currect depth (Computation $\rightarrow$ linear | act_function | identity would have $M_1 = 3$), and $\mathbb{1}_{x_i^j}\left(y_i^j\right)$ is the indicator function signaling whether $y_i^j$ is the same as $x_i^j$.

A brief analysis of the sensitivity of the distance calculation to the actual mutation costs has been carried out in Appendix D.

The **ValidPath** function is where the "Constrained" in "Constrained Smith-Waterman Crossover" comes from, discarding paths that would result in models that do not follow the grammar rules.

---

**Algorithm 6:** ValidPath

---

**Input:**    $path$, which holds the direction to reach each intermediate model
           $model1_{seq}$, the serialised representation of $model1$
           $model2_{seq}$, the serialised representation of $model2$
           $i$, the starting first index in the matrix
           $j$, the starting second index in the matrix
           $direction$, the direction we attempt to move towards from the starting position
**Output:** $valid$, which signals whether the direction would be a valid operation

```
1   node1 ← model1_seq[i]
2   node2 ← model2_seq[j]
3   if direction = "sub" then
4       if node1 and node2 are the same type of node then
5           if node1 and node2 are separators then
6               condition1 ← True
7               condition2 ← True
8               i ← i − 1
9               j ← j − 1
10              while condition1 or condition2 do
11                  if path[i,j] = "add" then
12                      i ← i − 1
13                  else if path[i,j] = "rem" then
14                      j ← j − 1
15                  else if path[i,j] = "mut" then
16                      i ← i − 1
17                      j ← j − 1
18                  if not condition1 then
19                      condition1 ← node1 is a separator that closes model1_seq[i]
20                  if not condition2 then
21                      condition2 ← node2 is a separator that closes model2_seq[j]
22              if path[i,j] = "mut" then valid ← True;
23              else valid ← False;
24          else valid ← True;
25      else valid ← False;
```

```
26  else if direction = "add" then
27  |   if node1 is a separator then
28  |   |   depth ← 0
29  |   |   i ← i − 1
30  |   |   while node1 does not close model1_seq[i] do
31  |   |   |   if model2_seq[j − i] is the start of a branching or a routing node then
32  |   |   |   |   depth_change ← 1
33  |   |   |   else if model2_seq[j − i] closes a branch or a routing node then
34  |   |   |   |   depth_change ← −1
35  |   |   |   else
36  |   |   |   |   depth_change ← 0
37  |   |   |   if path[i,j] = "add" then
38  |   |   |   |   i ← i − 1
39  |   |   |   else if path[i,j] = "rem" then
40  |   |   |   |   depth ← depth_change
41  |   |   |   |   j ← j − 1
42  |   |   |   else if path[i,j] = "mut" then
43  |   |   |   |   depth ← depth_change
44  |   |   |   |   i ← i − 1
45  |   |   |   |   j ← j − 1
46  |   |   if depth = 0 then valid ← True;
47  |   |   else valid ← False;
48  |   else valid ← True;
49  else if direction = "rem" then
50  |   if node2 is a separator then
51  |   |   depth ← 0
52  |   |   j ← j − 1
53  |   |   while node2 does not close model2_seq[j] do
54  |   |   |   if model1_seq[i − j] is the start of a branching or a routing node then
55  |   |   |   |   depth_change ← 1
56  |   |   |   else if model1_seq[i − j] closes a branch or a routing node then
57  |   |   |   |   depth_change ← −1
58  |   |   |   else
59  |   |   |   |   depth_change ← 0
60  |   |   |   if path[i,j] = "add" then
61  |   |   |   |   depth ← depth_change
62  |   |   |   |   i ← i − 1
63  |   |   |   else if path[i,j] = "rem" then
64  |   |   |   |   j ← j − 1
65  |   |   |   else if path[i,j] = "mut" then
66  |   |   |   |   depth ← depth_change
67  |   |   |   |   i ← i − 1
68  |   |   |   |   j ← j − 1
69  |   |   if depth = 0 then valid ← True;
70  |   |   else valid ← False;
71  |   else valid ← True;
```

In the case of attempting the substitution of one node into another, we first check that they are interchangeable. Substituting a Branching by a Computation, for instance, would result in an incongruous model, as the Computation node would not be suitable for holding the Branching node's children; thus, we only allow substitution between nodes of the same type, regardless of their hyperparameters. Note that Branching(2) is different from all other Branching nodes, as it holds four children instead of three, and thus are not interchangeable. If the nodes we want to substitute are both instances of the $node_{sep}$ class, we trace the path back to make sure that we substituted their associated opening nodes.

If we try to add or delete a module, we only have to be careful when dealing with a $node_{sep}$ instance. If that were the case, we ned to trace back the operations to ensure that (1) the associated opening node was dealt with with the same operation and (2) that we are not adding or deleting any non-closed Branching nor Routing nodes, nor any non-opened separator ones.

The **TraceBack** function is used to transform the $dists$ and $paths$ matrices into an actual set of operations that we can perform to transform $model1_{seq}$ into $model2_{seq}$.

---

**Algorithm 7:** TraceBack

---

**Input:**     $model1_{seq}$, the serialised representation of $model1$
           $model2_{seq}$, the serialised representation of $model2$
           $dists$, which holds the distance from $model1$ to every intermediate model
           $paths$, which holds the steps to reach each intermediate model from $model1$
**Output:** $ops_{valid}$, the list of operations to transform $model1$ into $model2$

---

1   $ops_{valid} \leftarrow \emptyset$
2   $i \leftarrow$ length of $model1_{seq}$
3   $j \leftarrow$ length of $model2_{seq}$
4   **while** $i > 0$ **or** $j > j$ **do**
5     $op_{new} \leftarrow \{$
6            $id \leftarrow$ length of $ops_{valid}$, serving as the operation identifier
7            $type \leftarrow$ the kind of operation ("add$_{n}ode$","$parallelise$","$substitute$"...)
8            $value \leftarrow$ the cost of performing the operation
9            $i \leftarrow$ the first index where the node starts
10           $j \leftarrow$ the second index where the node starts
11           $ii \leftarrow$, the first index where the node's separators start, if any
12           $jj \leftarrow$, the second index where the node's separators start, if any
13           $ops_{disabler} \leftarrow ops_{disabler}$, the operations that forbid performing $op_{new}$
14           $ops_{enabler} \leftarrow ops_{enabler}$, the operations that allow performing $op_{new}$
15           $\}$
16     **if** $op_{new}\{value\} > 0$ **then**
17        $ops_{valid} \leftarrow ops_{valid} \cup op_{new}$
18     **if** a branching or routing module has been added or deleted **then**
19        **for** all operations dealing with nodes contained within **do**
20           Update their disabler and enabler operations
21     **if** $path[i,j] = $ "add" **then**
22        $i \leftarrow i - 1$
23     **else if** $path[i,j] = $ "rem" **then**
24        $j \leftarrow j - 1$
25     **else if** $path[i,j] = $ "mut" **then**
26        $i \leftarrow i - 1$
27        $j \leftarrow j - 1$

---

Starting from the bottom right corner of the matrices, we follow the path until we reach the top left corner and save all the operations that have a cost, along with all the information required to perform them: what nodes are involved, their positions and, importantly, their interactions with the rest of the operations.

- Adding a branching or routing module around certain nodes is disabled by deleting all nodes inside, and enabled by adding any other node inside.

- Adding a branching or routing module and all nodes inside is disabled by itself, and enabled by adding any of the other inside nodes.
- Deleting a nodes inside a branching or routing is disabled by deleting all nodes inside said branching or routing, and enabled by either adding another node inside or deleting the branching or routing itself.

Knowing which operations enable and disable each other is crucial to be able to generate valid architectures that follow the grammar rules.

The **SelectOperations** function takes in all possible operations we can perform to substitute one parent model into another. It first validates each combination of operations, using their sets of disabler and enabler operations to check that no incongruous model would be created. It then assigns each combination a distance from the first model given by the value of each operation within. We then generate a truncated Gaussian probability distribution—optionally presenting a *skewness* parameter—that accommodates only values ranging from 0 to the maximum possible distance given by our operations. The probability of each combination of operations will then be drawn from the defined distribution and used to sample one $ops_{selected}$ at random out of all valid combinations.

By default, the *skewness* parameter is set to zero, generating a truncated, non-skewed Gaussian probability distribution. It could however be defined based on, for instance, the relative performance of both parents. This *skewness* parameter will make the *offspring* produced by $ops_{selected}$ resemble more closely the desired parent. Having a high skewness towards the best out of the two parents would enhance exploitation and reduce exploration, as the models generated would be closer to said parent. It might be interesting to increase this skewness as the search progresses and more promising regions in the space of architectures are found; however, considering that the exploitative RCSWX-based search strategies often underperformed when compared to exploration-focused approaches—see Figure 2—having a high skewness at the beginning would bias the search too much towards the first decent architectures found, hindering overall search performance. In any case, having a high skewness towards the worst performing parent would not make much sense, as it would induce higher exploitation of the least promising regions in the architecture space, benefiting from neither exploration nor thorough exploitation. The influence of the skewness parameter should be less noticeable the closer the parent models are, as these would belong to the same region of the architecture space and not have novel architectures to explore in between the parents.

---

**Algorithm 8:** SelectOperations

**Input:**    $ops_{valid}$, the set of possible operations
        $skewness$, the skewness of the operation sampling probability distribution
**Output:** $ops_{selected}$, a chosen subset of $ops_{valid}$

---

1  $combinations \leftarrow \emptyset$
2  **for** $i \leftarrow 1$ **to** (length of $ops_{valid}$)$^2$ **do**
3      $combo_{str} \leftarrow$ binary representation of $i$ with as many digits as operations in $ops_{valid}$
4      $ops \leftarrow$ a subset of $ops_{valid}$ given by $combo_{str}$
5      **if** $ops$ is a valid set of operations **then**
6         $combinations\{combo_{str}\} \leftarrow \sum ops_j\{value\}$
7  $distr \leftarrow$ a distribution with given $skewness$, truncated to 0 and $\max(combinations)$
8  $probs \leftarrow$ probability of each combination's value given by $distr$
9  $probs \leftarrow probs / \sum probs$
10  $combo_{selected} \leftarrow$ an operation subset from $combinations$ sampled with probability $probs$
11  $ops_{selected} \leftarrow$ a subset of $ops_{valid}$ given by $combo_{selected}$

---

## D   SENSITIVITY TO THE MUTATION COSTS

The distance metric—and thus, the subsequent search and loss landscape analysis—is completely dependent on the cost assigned to adding, removing or substituting layers in the input sequences. In order to better assess the influence of these costs, we have analysed the distance obtained when comparing 50 pairs of randomly generated models with the scoring matrices defined in Table 3.

Table 3: Scoring matrices defined

| Name | Addition/deletion cost | Mutation cost |
|---|---|---|
| Scoring matrix 0 | Always 1 | 0 if nodes are the same, 0.25 if nodes' children's hyperparameters differ, 0.5 if nodes' children's type of node differ and $\infty$ if parents type of node differ |
| S. M. 1 | Always 1 | 0 if nodes' children's are exactly the same, 0.5 if not, $\infty$ if parent's type of node differ |
| S. M. 2 | Always 1 | $\infty$ if parent's type of node differ, 0.5 otherwise |
| S. M. 3 | Number of branches for branching nodes, 1 otherwise | Difference in number of branches for branching nodes, same as default scoring matrix otherwise |

All models were sampled from *einspace*, having a length ranging from 4 to 104 layers, sampled uniformly. Each architecture was paired with another one with the same length. The results are depicted in the Figure 7.

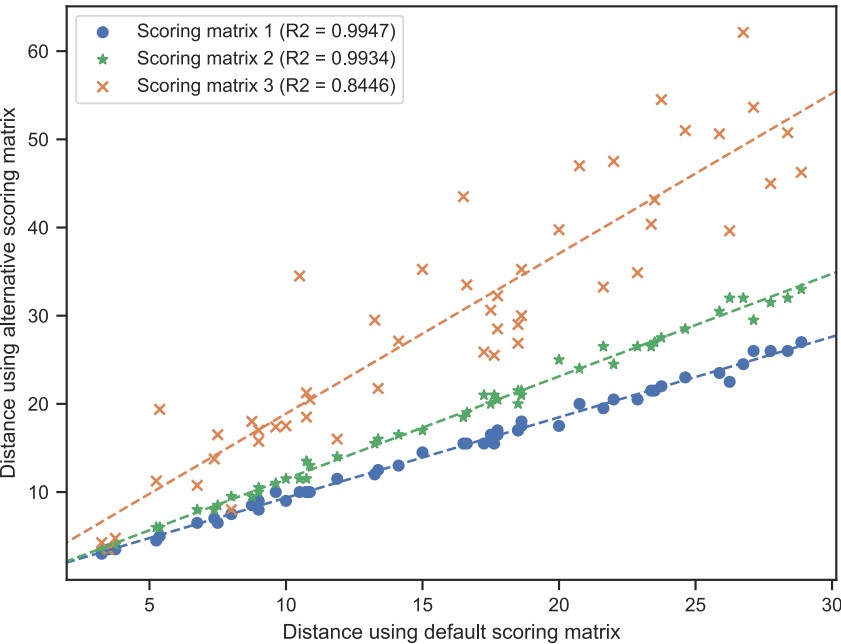

Figure 7: Distance calculated for randomly sampled pairs of models employing scoring matrices 0 (x axis) and 1 through 3 (y axis) with RCSWX, along with their linear fits and $R^2$.

It becomes clear that the scoring matrix changes the behaviour of the (R)CSWX because the relationship between the attained distances is not perfectly linear. The bigger the change in scoring matrix—in this case, weighting branching mutations by their number of branches, which can add a cost of 8 for a single operation rather than the maximum of 1 in the other scoring matrices—the higher the observed non-linearity. However, we can clearly see the correlation across the distances achieved, and thus we hypothesise that neither the loss landscape analysis nor the search results would be significantly changed by the choice of scoring matrix. Even if the loss landscape is warped, the clusters of architectures would remain close together, while distant architectures would remain distant. These differences would be the most noticeable for small-grain analysis and thorough exploitation of the space but, in those scenarios, the inherent randomness in the training—weight initialisation, data sampling, etc—would likely have a higher influence than a perturbation in the distance between models.

# E    CROSSING OVER PREDESIGNED MODELS

In this section, we explore the capabilities of RCSWX to compare and cross over relatively big predesigned models. We have constructed members of the ResNet and MLP-Mixer families in the *einspace* grammar and compared them, obtaining the following results.

Table 4: RCSWX results on the predefined architectures

| Architectures | number of nodes | compute time | distance |
|---|---|---|---|
| ResNet18, MLP-Mixer d8 | 267, 264 | 39.054 seconds | 60.38 |
| ResNet18, MLP-Mixer d12 | 267, 392 | 342.383 seconds | 95.38 |
| ResNet34, MLP-Mixer d8 | 499, 264 | 86.633 seconds | 119.62 |
| ResNet34, MLP-Mixer d12 | 499, 392 | 741.57 minutes | 115.62 |

The resulting alignment matrices are depicted in the following figures.

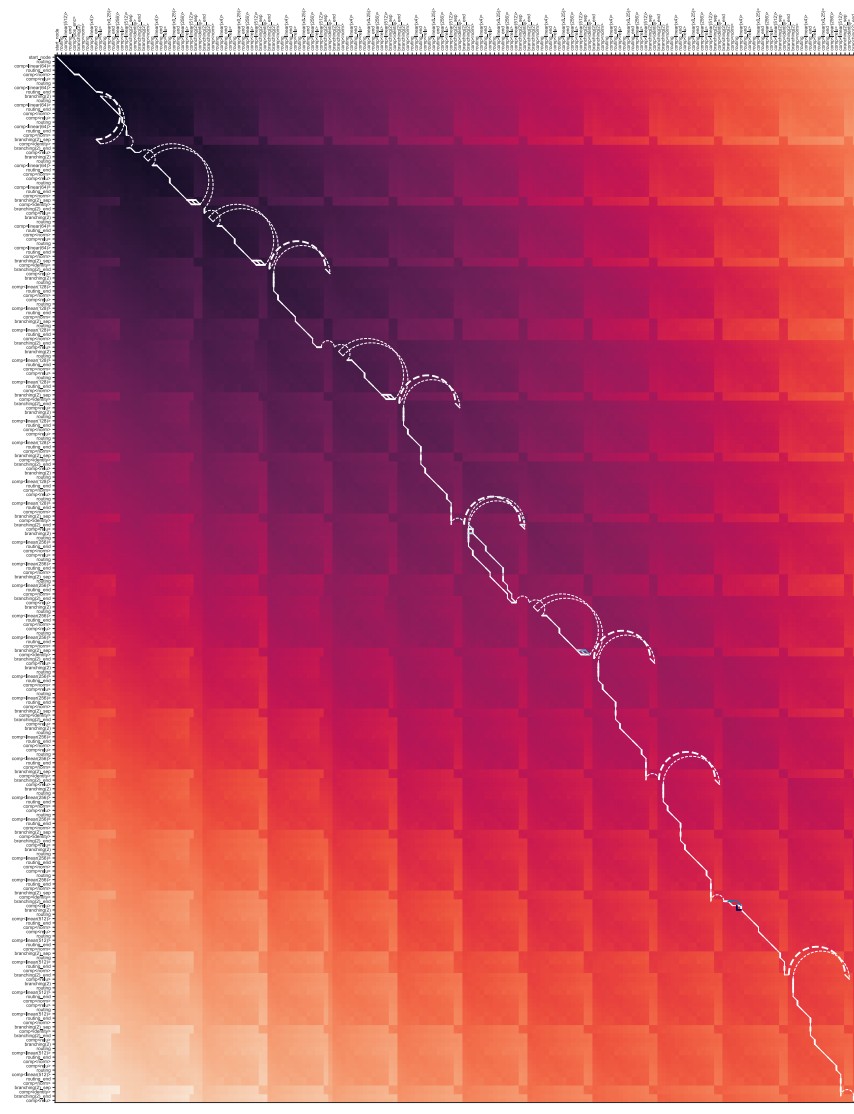

Figure 8: Alignment matrix for ResNet34 and MLP-Mixer d12

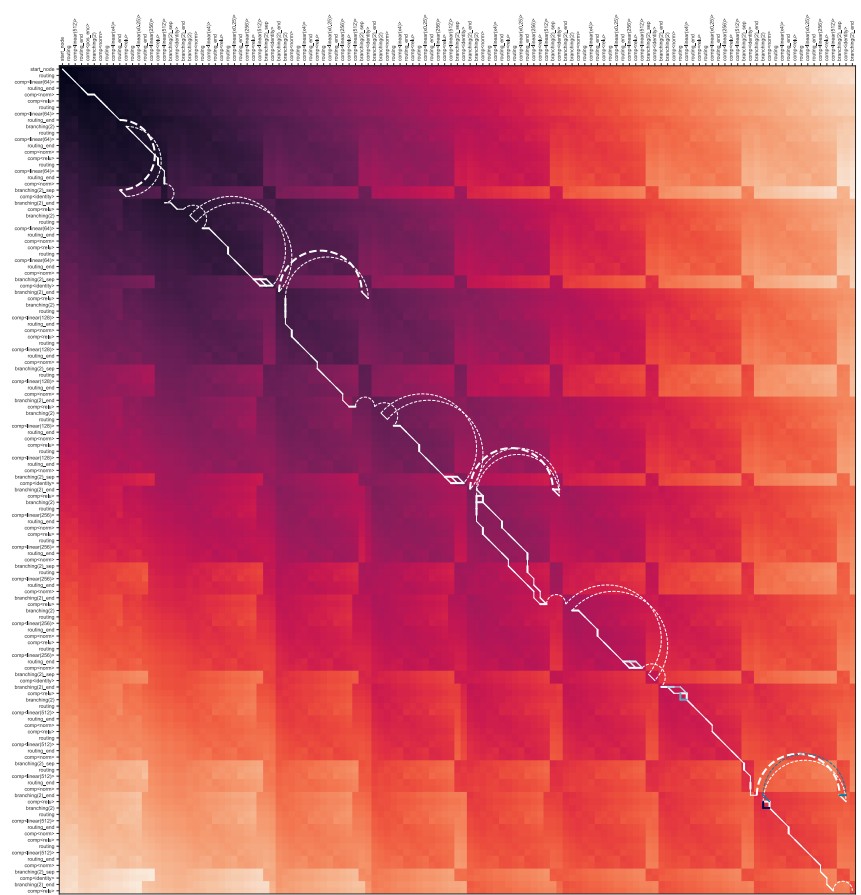

Figure 9: Alignment matrix for ResNet18 and MLP-Mixer d8

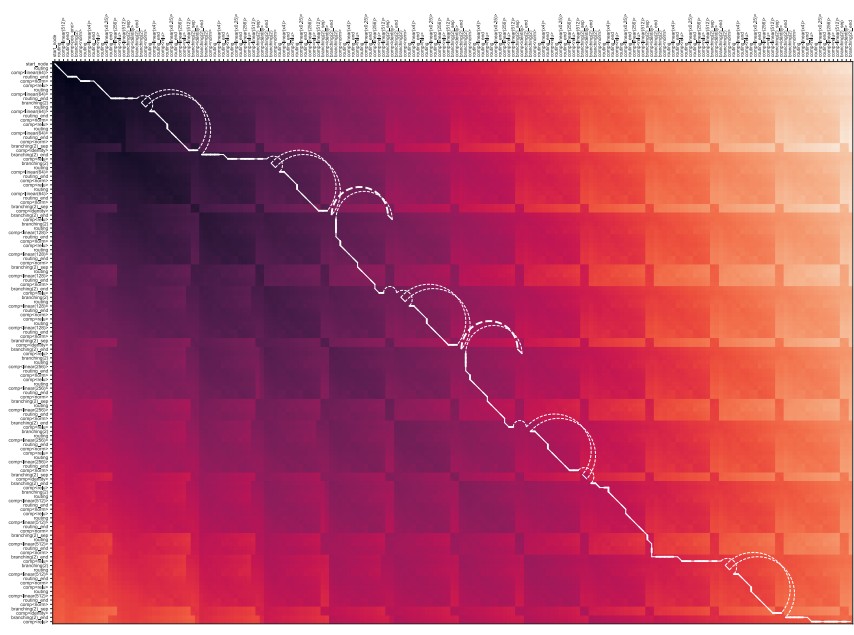

Figure 10: Alignment matrix for ResNet18 and MLP-Mixer d12

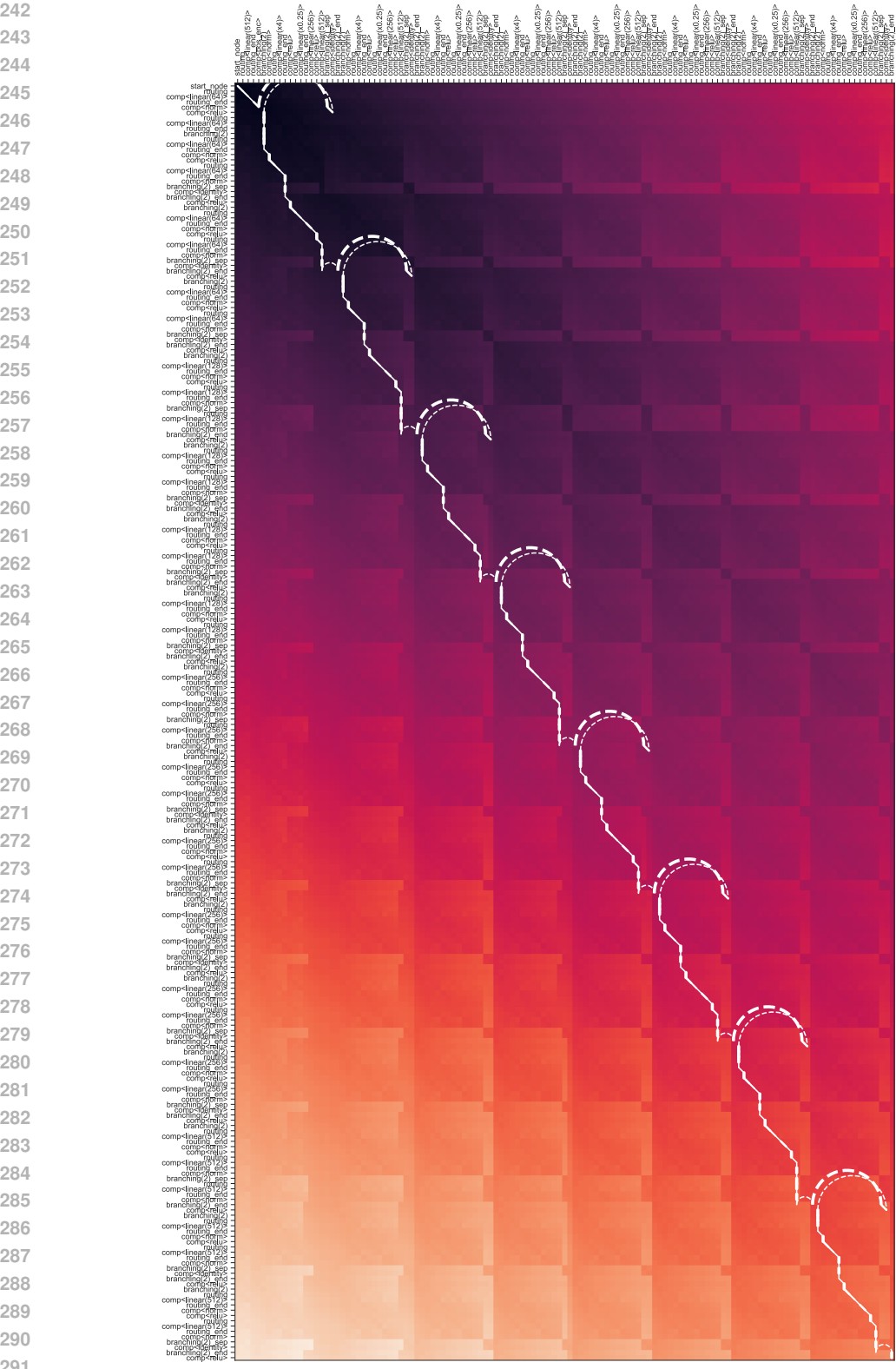

Figure 11: Alignment matrix for ResNet34 and MLP-Mixer d8

The repetitive block-based nature of the architectures becomes apparent when compared, with cyclical patterns—whose actual shapes and cycle lengths are defined by the alignment of the first and final layers of the architectures—spread throughout the shortest edit paths.

Also note that compute times are relatively low considering the large number of nodes for the aligment of ResNet18 and MLP-Mixer d8. This can be explained by the low depth throughout the matrix. There are no nested branches in either of the models, which lowers the amount of recursive computing—which translates into reduced computations, but also reduced overhead as well. However, compute times are relatively high for the alignment of ResNet34 and MLP-Mixer d12. This is due to redundant, irrelevant paths, mostly in the lower left and upper right corners, as checked empirically after the computations were performed. While alignments surrounding the shortest edit path are relatively constrained, there are a lot of equally low-performing paths further away towards the corners—e.g., it is as costly to mutate the first ResNet block into the first MLP-Mixer block and then add the second MLP-Mixer block than it is to add the first MLP-Mixer block and then mutate the first ResNet block into the second MLP-Mixer block. These paths are redundant in the sense that they achieve the same final distance and obey the same constraints. Keeping track of all these paths is not only memory intensive but compute intensive as well, as every possible path needs to be checked for constraints to continue forwards with the next operation.

The phenomena discussed above highlight the deviation in compute time from theory to practice and suggest that, for real applications with warm initialisations, RCSWX might be way faster than what is observed in Figure 3, given that the redundant and unpromising paths are collapsed or ignored. This collapse can either be cell-wise—each time we compute a cell, we collapse all redundant paths into one—or matrix-wise—considering how unlikely it is to find a best path very far away from the main diagonal of the alignment matrix, we can restrict the maximum number of path calculations at the corners by employing techniques akin to constraining with the Sakoe-Chiba band or the Itakura parallelogram (Geler et al., 2019).

## F  BROADER APPLICABILITY

In this work we have exemplified the use of (R)CSWX with the implementation of a simple genetic evolution optimiser to explore models expressed in the *einspace* grammar. However, our goal is to provide an efficient tool that enables researchers and NAS practitioners to deploy any crossover-based algorithm on their own search spaces, and even apply (R)CSWX to sequence alignment outside of the Deep Learning field. In this section, we suggest how to use (R)CSWX to construct different optimisers—using two or more parent architectures to generate offspring based both on model performance or relative distances—and work with complex search spaces to demonstrate its wide applicability.

### F.1  (R)CSWX ON COMPLEX SEARCH SPACES

#### F.1.1  MULTI INPUT-OUTPUT SPACES

Allowing multiple input and output nodes in an architecture can be achieved by paying with the constraints in the grammar—that is, by modifying the mutation costs. For spaces where all architectures present the same number of inputs and outputs, which is the most common case, it is enough to give the addition/deletion of input and output nodes—as well as the their substitution from or into any other type of node—a cost of $\infty$. This would force (R)CSWX to hinge around these nodes, aligning the intermediate segments. For cases with varying number of input and output nodes, input and output nodes can be treated as any other type of node as there should not be any actual constraints on their position within the architectures. If we want the best paths to align as many of these nodes as possible, though, it would be sufficient to set the cost of adding/deleting these nodes to a high enough number—for instance, if adding or removing any other node has a cost of 1, the cost of adding or removing an input/output node can simply be the length of the biggest parent—and the cost of substituting from or into any other type to $\infty$. An example alignment matrix calculated in this fashion is shown in Figure 12 below.

Note that the case with aligned input and output nodes can be treated as a specific, fortuitous, instance of the unaligned multi-input and -output; however, setting the addition/deletion costs to $\infty$ instead of a large value allows to directly flag and ignore all positions in the alignment matrix outside the regions defined by these hinge points, speeding up computations considerably for large models.

| | | START | | B | | INPUT | | A | | A | | INPUT | | A | | OUTPUT | |
|---|---|---|---|---|---|---|---|---|---|---|---|---|---|---|---|---|---|
| START | | 0 | 0 | ∞ | ∞ | ∞ | ∞ | ∞ | ∞ | ∞ | ∞ | ∞ | ∞ | ∞ | ∞ | ∞ | ∞ |
| | | 0 | 0 | 0+1 | 1 | 1+8 | 9 | 9+1 | 10 | 10+1 | 11 | 11+8 | 19 | 19+1 | 20 | 20+1 | 21 |
| A | | ∞ | 0+1 | 0+0.5 | 1+1 | ∞ | 9+1 | 9+0 | 10+1 | 10+0 | 11+1 | ∞ | 19+1 | 19+0 | 20+1 | 20+0.5 | 21+1 |
| | | ∞ | 1 | 1+1 | 0.5 | 0.5+8 | 8.5 | 8.5+1 | 9 | 9+1 | 10 | 10+8 | 18 | 18+1 | 19 | 19+1 | 20 |
| A | | ∞ | 1+1 | 1+0.5 | 0.5+1 | ∞ | 8.5+1 | 8.5+0 | 9+1 | 9+0 | 10+1 | ∞ | 18+1 | 18+0 | 19+1 | 19+0.5 | 20+1 |
| | | ∞ | 2 | 2+1 | 1.5 | 1.5+8 | 9.5 | 9.5+1 | 8.5 | 8.5+1 | 9 | 9+8 | 17 | 17+1 | 18 | 18+1 | 19 |
| INPUT | | ∞ | 2+8 | ∞ | 1.5+8 | 1.5+0 | 9.5+8 | ∞ | 8.5+8 | ∞ | 9+8 | 9+0 | 17+8 | ∞ | 18+8 | ∞ | 19+8 |
| | | ∞ | 10 | 10+1 | 9.5 | 9.5+8 | 1.5 | 1.5+1 | 2.5 | 2.5+1 | 3.5 | 3.5+8 | 9 | 9+1 | 10 | 10+1 | 11 |
| B | | ∞ | 10+1 | 10+0 | 9.5+1 | ∞ | 1.5+1 | 1.5+0.5 | 2.5+1 | 2.5+0.5 | 3.5+1 | ∞ | 9+1 | 9+0.5 | 10+1 | 10+0 | 11+1 |
| | | ∞ | 11 | 11+1 | 10 | 10+8 | 2.5 | 2.5+1 | 2 | 2+1 | 3 | 3+8 | 10 | 10+1 | 9.5 | 9.5+1 | 10 |
| A | | ∞ | 11+1 | 11+0.5 | 10+1 | ∞ | 2.5+1 | 2.5+0 | 2+1 | 2+0 | 3+1 | ∞ | 10+1 | 10+0 | 9.5+1 | 9.5+0.5 | 10+1 |
| | | ∞ | 12 | 12+1 | 11 | 11+8 | 3.5 | 3.5+1 | 2.5 | 2.5+1 | 2 | 2+8 | 10 | 10+1 | 10 | 10+1 | 10 |
| OUTPUT | | ∞ | 12+8 | ∞ | 11+8 | ∞ | 3.5+8 | ∞ | 2.5+8 | ∞ | 2+8 | ∞ | 10+8 | ∞ | 10+8 | 10+0 | 10+8 |
| | | ∞ | 20 | 20+1 | 19 | 19+8 | 11.5 | 11.5+1 | 10.5 | 10.5+1 | 10 | 10+8 | 18 | 18+1 | 18 | 18+8 | 10 |

Figure 12: Alignment matrix for two sequences with a differing number of input nodes (addition/deletion cost of 1; mutation cost of 0 if nodes are the same, 0.5 if they are interchangeable). Each matrix cell is subdivided into substitution, deletion, addition and selected costs. Selected operations are shown in a light colour, and deprecated operations are shown in grey. The shortest edit path is shown in green.

### F.1.2 RECURSIVE GRAPHS

Aligning recursive architectures using the (R)CSWX is not a trivial task. The best way to approach this is to make sure that the sequential representation of the graph is representative, reversible and unique—or, at least, limited in such a way that we can calculate all possible alignments recursively in a reasonable amount of time—and, then, define the mutation costs appropriately to make sure that the constraints of the grammar are respected. Using the input and output nodes as anchor points, we can define the sequence of nodes employing simple graph traversal algorithms, like breadth-first search, depth-first search, or encoder-based approaches, like (Xu et al., 2018; Liu and Ji, 2022; Chen et al., 2025). Special separator tokens can be introduced during the definition of the sequences. By carefully defining the addition, deletion and substitution costs for these separator tokens, we can force the shortest edit paths to respect the rules of the space we are exploring—similarly to what has been accomplished with the constrains imposed to the `Branching(2)` layers in the *einspace* grammar—and/or weight the alignment of the architecture loops according to our needs.

### F.2 (R)CSWX TO DEFINE SEARCH ALGORITHMS

### F.2.1 PARTICLE SWARM OPTIMISATION

In particle swarm optimisation (PSO) (Kennedy and Eberhart, 1995), each individual in the population calculates its new position in the space by moving in a direction defined by a vector, which is interpolated from the vector pointing towards the best position achieved by the individual so far and the vector pointing towards the best individual in the current population. These two components can be weighted by the fitness of their respective individuals, and the interpolated vector can be rescaled

based on a given step size. This approach has already been successfully applied to NAS (Lankford and Grimes, 2024; Turky et al., 2026), but its application is heavily hindered by the ability to perform crossovers in the search space defined. We propose a method to perform multi-parent crossover by pooling the mutation operations yielded by (R)CSWX into four sets from which to sample from, allowing the deployment of any multi-parent interpolation-based algorithm, like PSO.

First, the original model is aligned with both parents separately using (R)CSWX. Then, the two resulting sets of operations are compared with one another. Those pairs of operations that are exactly the same—e.g., removing a `Computation(ReLU)` node from the original individual that is not present in either one of the other two parents—are bagged together in set $A$. Then, those pairs of operations that are similar—e.g., mutating a `Computation(ReLU)` node into a `Computation(linear64)` in one of the alignments and into a `Computation(linear32)` in another—are bagged together into set $B$. The rest of the operations are bagged individually in sets $C_1$ and $C_2$, according to the alignment matrix they belong to.

The number—or combine cost—of operations to be performed can be modulated by a step size parameter. We sample operations at random from set $A$ until needed. If we sample all operations from set $A$, we continue sampling pairs of operations from set $B$. The actual operation to perform out of the pair can be sampled according to the individuals' fitness scores. If set $B$ is depleted, we select operations from the remaining sets $C_1$ and $C_2$. Once again, the set from which to sample each operation can be selected probabilistically based on the individuals' respective fitness.

After applying the selected operations, the generated offspring is guaranteed to be closer to both parents simultaneously, given that they share some common nodes. This method can be expanded to perform multi-crossover across any number of parents, but the probability of finding common operations reduces drastically and the definition of further operation sets—e.g., operations shared by all alignments, operations shared by all but one alignments, etc—might be necessary to avoid treating all operations as unpaired.

### F.2.2 ESTIMATION OF DISTRIBUTION ALGORITHMS

Using the distances across all models in the population, we can define a high dimensional space—similarly to Section 5.3—and construct a probabilistic model that shifts the sampling of the new offspring towards the most promising regions of the space. This sampling can be performed by either (1) selecting parents for crossover that define lines that cross over said promising regions and using the skewness parameter to control the sampling while remaining probabilistic or (2) performing multi-parent interpolation similarly to the approach described to perform PSO.

### F.2.3 DIFFERENTIAL EVOLUTION

Differential Evolution (DE) can be implemented by combining the sets of operations yielded by (R)CSWX. A population size $NP \geq 4$ and a crossover probability $CR \in [0,1]$ need to be selected. Due to the space to explore being combinatorial rather than numerical, the definition of a differential weight $F \in [0,2]$ is not trivial and we opt to simply set it to 1.

First, for each individual in the population $x_n$, we select three other individuals $x_1$, $x_2$ and $x_3$, and define the set of operations that transform models $x_a$ and $x_b$ as $\Omega_{x_a, x_b} = x_a - x_b$. We can then calculate a donor set of operations $\Omega_{\text{donor}}$ as $\Omega_{\text{donor}} = (x_1 - x_n) \cup ((x_2 - x_n) \notin (x_3 - x_n))$. A single operation is sampled out of $\Omega_{\text{donor}}$ to mimic the behaviour of the $\delta$ random variable in the DE algorithm. Then, for each other operation in $\Omega_{\text{donor}}$, we discard it with probability $p = 1 - CR$, generating the set $\Omega_{\text{final}}$. To generate the new individuals, we simply apply all operations in $\Omega_{\text{final}}$ to $x_n$.

### F.2.4 FIREFLY ALGORITHM

In the firefly algorithm, for each search step $t$, each individual $x_i$ in the population of size $NP$ is updated by moving towards each other individual $x_j$ that presents a better fitness based on their distance $r_{ij}$ and a random permutation $\alpha_t \epsilon_t$ such that $x_i^{t+1} = x_i^t + \alpha_t \epsilon_t + \sum_{j=1}^{NP} \beta(x_j^t - x_i^t) e^{-\gamma r_{ij}^2}$. The random permutation can be achieved with a simple mutation, while the term that moves each individual towards the best ones in the population can be calculated using (R)CSWX.

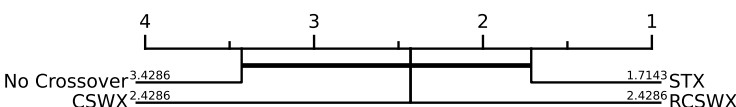

Figure 13: Critical difference (CD) diagram comparing the average ranks of the four algorithms across all datasets. The Friedman test did not reveal a statistically significant difference among the methods (p = 0.0997), and none of the Wilcoxon–Holm post-hoc comparisons reached significance. All algorithms fall within a single cluster—indicated by the horizontal connection—showing that their rank differences do not exceed the critical threshold.

First, for each individual $x_i$ we select every architecture $x_j$ that presents better fitness and calculate the set of operations $\Omega_j$ that transforms $x_i$ into $x_j$. Then, for each operation across all models that pertain to a certain node—e.g., one node in $x_i$ is a `Computation(linear64)` that is either removed or changed to a different type of `Computation` node in some of the alignments—we select the one pertaining to the alignment with individual $x_j$ with a probability proportional to $e^{-\gamma r_{ij}^2}$, generating a set of operations $\Omega_{\text{selected}}$. These distances $r_{ij}$ are already known as it is given by (R)CSWX, which has already been used to generate the sets of mutation operations. Lastly, each operation in $\Omega_{\text{selected}}$ is either selected or discarded based off a probability $\beta$, generating $\Omega_{\text{final}}$. By applying the operations in $\Omega_{\text{final}}$, we produce an offspring architecture that is interpolated from the population based on both their relative fitnesses and distances.

### F.2.5 CUSTOM DIVERSITY-DRIVEN ALGORITHMS

The distance metric provided by RCSWX can be employed to explicitly control the exploration–exploitation tradeoff, be it by regularising architecture scores with a diversity term or selecting a novel individual for training out of a batch of mutated ones. This allows for tweaking already existing optimisers or even proposing new ones. Explicitly controlling the diversity of the population is not only expected to further improve results by helping escape local minima, but is also crucial for deep ensemble learning and other scenarios where non-regularised NAS tends to underperform.

## G  SIGNIFICANCE TESTING

To assess whether the four algorithms—mutation-only, STX, CSWX and RCSWX-driven evolutionary algorithms—differ significantly in performance across datasets, we applied a non-parametric Friedman test to their accuracy ranks in the datasets' test partitions. The statistical test did not reject the null hypothesis that all algorithms perform equivalently (p = 0.0997), indicating that the observed differences in average rank are not statistically meaningful. For completeness, we also conducted Wilcoxon signed-rank pairwise comparisons with Holm correction, with none of the pairwise comparisons recognised as significant. The corresponding critical difference (CD) diagram reflects this outcome: all algorithms fall within a single cluster, illustrating that their rank differences do not exceed the critical threshold. These results together indicate that we cannot conclude that any algorithm outperforms the others across the evaluated datasets.

## LLM USAGE

Large language models were used only to aid and polish the writing of this paper, as well as auto-complete code fragments.

