# OpenReview forum: "Evolutionary Architecture Search Through Grammar-based Sequence Alignment"
_ICLR.cc/2026/Conference — Submitted to ICLR 2026_

### Official Review · Reviewer_JjPP · 2025-10-22

**Soundness:** 2
**Presentation:** 2
**Contribution:** 2
**Rating:** 2
**Confidence:** 4

**Summary:**

In this paper, authors introduce constrained and recursive Smith-Waterman style sequence alignment algorithms, CSWX and RCSWX, as efficient, grammar-based tools for NAS applications in expressive grammar-defined spaces. The proposed methods serve as both architecture distance metrics and as crossover operators for evolutionary-population-based search. The authors claim to vastly reduce the computational complexity of crossover and distance computation compared to prior methods, and retaining or even improving search efficiency. Applications shown include efficient diversity measurement, loss landscape analysis, and practical evolutionary search in large grammar-based NAS benchmarks, with empirical results supporting the methods’ computational scalability and competitive search performance.

**Strengths:**

1. The adaption of constrained Smith-Waterman alignment to grammar-based NAS is both novel and well-motivated. The recursive variant to handle permutation invariance demonstrates sophisticated algorithmic thinking.

2. The formal analysis on the metric properties, runtime complexity, and permutation invariance, provide relatively solid theoretical foundations.

3. The experiments span multiple dimensions, e.g., search performance, scalability, landscape analysis, offer clear and quantitative head-to-head comparisons across five datasets and several search strategies, e.g., STX, SEPX and no crossover.

**Weaknesses:**

1. While the authors acknowledge their focus is on introducing tools rather than achieving SoTA performance, the experiments use only 1000 architecture evaluations and relatively small datasets, e.g., CIFAR-10. Besides, there is no direct comparison with the most recent SoTA NAS approaches beyond the baselines. These weakens the empirical claim of broader impact and applicability.

2. As shown in Table 1, the proposed RCSWX shows less robust results compared to CSWX, specifically, with lower mean accuracy and higher variance on some benchmarks. The authors’ explanation of the difference between 'perfect interpolation' and 'noise injection' is plausible but lacks rigorous analysis. Additionally, only three random seeds are used, which is limited for high-variance evolutionary methods, and statistical significance is not analysed.

3. Another critical weakness is the absent of ablation studies for critical scoring function choices, e.g., the cost matrix or arbitrary constants in SubstitutionCost. I wonder how sensitive are performance and search landscape properties to these choices? Additionally, the selection function and skewness hyper-parameter’s  impact is not explored.

4. While the proposed methods are well-motivated for grammar-based spaces, they are only shown for the 'einspace' family and similar grammars. Potential obstacles for spaces with more irregular grammar rules, complex parameter sharing, or multi-input/-output components are not discussed. Authors claim 'any search space that can be represented as a sequential set of tokens', but there is not critically assessed. These make the extensibility and generalisation of proposed methods more like an assertion.

**Questions:**

1. What’s the impact of different choices for the cost matrix in SubstitutionCost?

2. Given the high variance in the evolutionary search, why are only three seeds used? Have you conduct significant testing?

3. Can the proposed methods be readily adapted for use in non-evolutionary NAS?

---

> ### Author Response · Authors · 2025-11-21
>
> We are flattered to hear read your identified strengths regarding the novelty and soundness of our algorithms, as well as the extension of our experiments and analysis. We thank you for the thorough feedback, which has led to substantial changes to the paper. Regarding the specific concerns expressed in your review:
>
> W1: We have added Appendix F in order to substantiate our claim that (R)CSWX is widely applicable, and thus can be a good stepping stone towards the advancement of better NAS search algorithms. Moreover, we are working on extending our experiments to span more datasets and seeds to provide a more statistically sound analysis of the behavior of the different operators in the evolutionary algorithm scenario, and will extend the discussion to reflect upon the observed behaviour.
> W2: We are working on extending our experiments to span more datasets and seeds to provide a more statistically sound analysis of the behavior of the different operators in the simple evolutionary algorithm scenario. Moreover, we are performing the loss landscape analysis in a second dataset to exemplify how they change across distinct data distributions, which serves to substantiate our claims on the need for controlling the exploration-exploitation trade-off and thus explain the differences observed in the performance of the different crossover operators. The discussion section has been updated accordingly.
> W3: We have addressed the influence of the scoring procedure in the new Appendix D. We have also expanded the discussion on the properties of the skewness hyperparameter in Appendix C. Unfortunately, a proper empirical characterisation of the influence of these hyperparameters would require significant experimental additions, and we simply do not have the compute necessary to prioritise this.
> W4: We stand by the claim that samples from any space that can be represented as a sequential set of tokens can be compared and crossed over with (R)CSWX. As long as the mutation costs are properly defined to ensure the grammar constraints, (R)CSWX will serve as a proper distance metric and compute shortest edit path(s). In order to exemplify the use of (R)CSWX to more grammars, Appendix F has been added where we explore how to define different constraints for complex grammars.
> Q1: We have addressed this in the new Appendix D.
> Q2: At the time of submission, only three seeds had completed running for all the methods depicted. We are currently working on augmenting the number of datasets to 7 and seeds to 5 in order to compensate for the inherent randomness in both model generation and training.
> Q3: We have addressed the use of (R)CSWX for different search strategies in Appendix F, showing that, indeed, the tools provided can be employed without any changes to construct optimizers outside of the evolutionary NAS realm.
>
> We hope that these answers are sufficient to address your concerns. We would appreciate a reconsideration of the score, provided that you found the changes performed to the paper satisfactory. We would also love to hear if you have any further questions or concerns, and thank you again for your thorough review.

---

> > ### Comment · Reviewer_JjPP · 2025-11-27
> >
> > I'd like to thank the authors for their response and clarifications. However,
> > 1) the current version of the manuscript is still lacking in comparison with SoTA NAS methods, although the authors claimed that they're working on expanding the experiments;
> > 2) the authors responded with verbal discussion for W2 but no rigorous theoretical or empirical analysis;
> > 3) the authors added Appendix D with a small sensitivity test on cost matrices, but nothing is added for skewness or selection function;
> > 4) the updated Appendix F is basically conceptual, but no experiments or actual complex grammar benchmark provided.
> >
> > Due to the above reasons, as well as considering points from other reviewers, I lean to maintain my initial rating.

---

### Official Review · Reviewer_hK6c · 2025-10-27

**Soundness:** 2
**Presentation:** 2
**Contribution:** 2
**Rating:** 4
**Confidence:** 3

**Summary:**

This paper introduces two algorithms (CSWX and RCSWX) based on the Smith-Waterman sequence alignment method, to enable efficient evolutionary search in expressive, grammar-based Neural Architecture Search (NAS) spaces. These methods provide a computationally tractable way to calculate an edit distance between neural architectures. This distance is then leveraged not only as a crossover operator to create hybrid offspring, but also as a formal distance metric to analyze the architectural loss landscape and population diversity. The authors demonstrate experimentally that their approach is orders of magnitude faster than prior graph-edit-distance methods and achieves competitive performance in evolutionary searches.

**Strengths:**

+ The paper clearly shows that prior methods based on Graph Edit Distance (e.g., SEPX) are NP-hard and become intractable for even moderately sized graphs. The proposed (R)CSWX methods, by contrast, are highly efficient, effectively scaling to large architectures. This is a significant practical contribution.

+ The proposed method is valuable as both a search operator and an analysis tool. Using $d_{RCSWX}$ to perform a large-scale analysis of the architectural loss landscape may be a compelling application.

+ The paper is well-written and clearly structured.

**Weaknesses:**

+ As is well known, the primary computational cost of NAS is the performance estimation (i.e., architecture evaluation), not the search strategy. Therefore, the main contribution of this paper, which enhances the efficiency of the search strategy, seems less significant in the context of the overall NAS pipeline.

+ In Table 1, RCSWX shows the lowest average performance, even underperforming the "No Crossover" baseline. Its result on AddNIST, in particular, is very poor. This undermines the central motivation for developing the more complex, permutation-invariant operator.

+ It would have been highly valuable to include a comparison on a smaller benchmark where SEPX is tractable (e.g., NAS-Bench-101). This would have established whether (R)CSWX, which approximates the edit path, produces offspring of comparable or superior quality to the true shortest edit path crossover.

+ The "skewness" parameter (Algorithm 1) appears to be important for guiding the sampling of operations. The paper mentions it can be set based on parent performance, but it is not discussed in the experimental setup. The sensitivity of the search to this parameter is unknown.

**Questions:**

+ Given the poor search performance of RCSWX compared to the simpler CSWX, could the author elaborate on the hypothesis that "perfect interpolation" is detrimental?

+ How was the "skewness" parameter for operation sampling set during the search performance experiments in Section 5.1? How sensitive is the performance of CSWX and RCSWX to this hyperparameter?

---

> ### Author Response · Authors · 2025-11-21
>
> We are really glad that you appreciated both the algorithmic efficiency and the proposed applications, as well as the structure and clarity of the paper. We wholeheartedly appreciate your feedback, and hope that we address your questions in the following answers:
>
> W1: That is true for most works in the literature, which either focus on simpler, more constrained search spaces where implementing a crossover operator is trivial, or simply do not employ crossover at all. As we show both theoretically and empirically, exact crossover methods such as SEPX are completely intractable in large search spaces and would take much longer than the actual performance estimation when crossing over even moderately sized architectures. Our method brings the compute required by crossover-based search strategies down to the one required for the aforementioned simpler spaces. Moreover, since crossover has been shown to improve performance in simpler search spaces [1,2], we expect it to also provide benefits in huge search spaces where the exploration-exploitation trade-off becomes much more important.
> W2: We have expanded the discussion section to characterise the behaviour of the different methods, at least as observed within the simple evolutionary search strategy employed. This discussion will serve to select or even propose more performant search methods (like the ones described in the new Appendix F) in future works. Even if RCSWX was proven to underperform independent of the search strategy, it is still a crucial tool for controlling exploration and analysing the architectural loss landscapes, which is an important yet still unexplored topic in NAS.
> W3: SEPX and RCSWX give the exact same list of operations when mutating one parent into another (they both output the true shortest edit path) and they only differ in their computational complexity. We have confirmed this empirically and expanded Section 5.2 accordingly. As such, if we used the same operation sampling strategy, the search results on a smaller space like NAS-Bench-101 would be identical using SEPX or RCSWX and therefore not add anything meaningful to the experimental analysis.
> W4: We have expanded the discussion on the properties of the skewness hyperparameter in Appendix C. While it can theoretically be used to control the exploration-exploitation tradeoff, for all experiments described in the paper, it has been set to zero for simplicity, generating truncated non-skewed Gaussian probability distributions to sample the operations from. We have added discussion on this in Appendix C.
> Q1: We have expanded the discussion section to expand on the properties of permutation sensitive and permutation invariant approaches and how they translate into varying convergence speeds and performances across loss landscapes.
> Q2: The skewness parameter, as well as the substitution costs and other (R)CSWX hyperparameters, were set to their default values depicted in Appendix C. The sensitiveness of the (R)CSWX distance metric to the mutation costs defined has been explored in the new Appendix D. Higher skewness would entail higher exploitation and lower exploration during the search, but we hypothesise that the loss landscape analysis would not be significantly affected.
>
> If you have any further concerns, please do not hesitate to express them! We would also appreciate a reassessment of the score given the changes made to the paper.
>
> [1] He, C., Tan, H., Huang, S., & Cheng, R. (2021). Efficient evolutionary neural architecture search by modular inheritable crossover. Swarm and Evolutionary Computation, 64, 100894.
> [2] Qiu, X., & Miikkulainen, R. (2022). Shortest Edit Path Crossover: A Theory-driven Solution to the Permutation Problem in Evolutionary Neural Architecture Search (Version 4). arXiv.

---

### Official Review · Reviewer_4JE4 · 2025-10-31

**Soundness:** 3
**Presentation:** 4
**Contribution:** 2
**Rating:** 4
**Confidence:** 4

**Summary:**

This paper introduces (R)CSWX, an algorithm for computing distances and performing crossover in grammar-based NAS search spaces, based on the Smith-Waterman algorithm. The proposed algorithms scales well to large and complex spaces, and CSWX performs well compared to the baseline (no crossover) but RCSWX offers no empirical improvement on average.

**Strengths:**

1. (R)CSWX is an interesting algorithm, building on the Smith-Waterman algorithm, for NAS.
2. Runtime analysis presents good upper bound computational complexity.

**Weaknesses:**

W1. RCSWX which is the more tractable and practical algorithm underperforms the baselines.
W2. The authors offer insufficient analysis or explanation for W1.
W3. The authors state that the focus of this work is to "introduce a theoretically sound, computationally efficient crossover operator for grammar-based NAS, intended as a tool for further research rather than as a benchmark for state-of-the-art performance." but this is not a sufficient reason to not compare CSWX and RCSWX with other optimisation strategies .
W4. In light of W3, the experiments were also lacking, with no comparison to similar NAS algorithms.

**Questions:**

1. How does Figure 2 demonstrate good exploration/exploration trade-off?
2. How does CSWX empirically scale further than 200 nodes?

---

> ### Author Response · Authors · 2025-11-21
>
> We really appreciate your feedback, and we are very pleased to hear that you appreciated the algorithmic novelty and increase in computational efficiency we provide. We are eager to further exploit the capabilities of the proposed methods, both as a search and as an analysis tool. Regarding the identified weaknesses and questions:
>
> W1: While the evolutionary search method employed seems to benefit from the emergent properties of “imperfect” alignment options like STX or CSWX, the proposed RCSWX outperforms all baselines in terms of shortest edit path calculation speed without sacrificing accuracy. Most of the information provided by (R)CSWX, such as edit distance or element-wise alignment cost, can be employed to construct better performing optimisers and is out of scope of this work, which only focuses on the proposal and characterisation of the crossover/distance metric tool itself and instantiates it inside a standard evolutionary algorithm. Further work will focus on exploiting its potential for exploration (as suggested in Appendix F), as well as expand upon the loss landscape analysis to better characterise how to explore NAS spaces.
> W2: We have expanded the discussion section to provide explanations for the observed behaviours, as well as future research directions to confirm our hypotheses. Moreover, we are working on strengthening our claims through further loss landscape analysis on a second dataset.
> W3: It is true that the simple evolutionary search itself has not been compared to other search strategies in these architecture loss landscapes. However, the (R)CSWX algorithms are not optimisation strategies but rather crossover operators, and have been compared to the most performant available methods for model crossover and edit distance computation in the literature, proving both theoretically and empirically to produce the same offspring in much less time (Section 5.2).
> W4. Unfortunately, we lack the compute to perform a comprehensive analysis on the performance of (R)CSWX when combined with distinct optimisation strategies, which were previously unfeasible to be deployed in complex architectural spaces. We have however added Appendix F to encourage further research on this topic. Moreover, we are working on expanding our analysis to 7 datasets and 5 seeds to provide a more statistically significant set of results for the search strategy that we do consider.
> Q1: Figure 2 does not demonstrate good exploration/exploitation trade-off but, rather, the need for one. Exploration-based search methods thrive in some datasets where exploitation-intensive strategies fail to escape local regions, but are unable to properly converge in datasets where good regions are easy to find and difficult to optimize. The proposed RCSWX, as a distance metric, opens up the possibility of analysing the loss landscape and defining better strategies to explore it and, as a crossover operator, enables the definition of any interpolation-based search strategy, as explored in the new Appendix F.
> Q2: We have provided the alignment matrix for big models (264 to 499 nodes in size) along with compute times and distance scores in Appendix E, along with some insights on the performance of the RCSWX in real world scenarios.
>
> We hope that you find both these answers and the changes performed to the paper satisfactory, and would appreciate a reconsideration of the score given. Please do not hesitate to express any further concerns regarding our contribution.

---

### Official Review · Reviewer_z5eP · 2025-11-03

**Soundness:** 3
**Presentation:** 2
**Contribution:** 2
**Rating:** 4
**Confidence:** 4

**Summary:**

This paper introduces two variants of the Smith-Waterman (i.e., Constrained Smith–Waterman Crossover - CSWX, and recursive CSWX) to compute the edit distance in grammar-based evolutionary architecture search, and perform crossover for the selected parent. CSWX first converts each parent architecture into a simplified sequence, then computes the minimum-cost alignment path between them, and finally generates an offspring along that path. Experiments are performed on Unseen NAS benchmark, demonstrating the effectiveness of CSWX and RCSWX.

**Strengths:**

- The proposed variants seem reasonable.
- As shown in Table 1, crossover shows better performance than mutation only.

**Weaknesses:**

- As stated in the conclusion, CSWX sometimes outperforms RCSWX,  which requires a deep investigation.
- In Figure 2, no crossover shows better validation performance on Chesseract and Isabella, but achieves lower test performance on test set as shown in Table 1. It is unclear why this happens.
- The performance gain is not consistent across datasets, which raises concerns about the stability of the method when applied to different datasets. Sometimes CSWX does not better than STX (see Chesseract and MulTNIST in Table 1).

**Questions:**

My main concern about the paper is the experimental results. The performance gain is not significant (compared to STX) and is sometimes not stable across datasets.

---

> ### Author Response · Authors · 2025-11-21
>
> We thank you for your feedback, and we hope that you find the changes we made to the paper, as well as the answers provided to the identified weaknesses and questions, satisfactory.
>
> W1: We have expanded the discussion section to delve deeper into the properties of permutation sensitive and permutation invariant interpolation, and how CSWX can introduce both heavier exploration (due to greater offspring complexity changes during operation subsampling) and a kind of structured mutation, which may be beneficial for some loss landscapes. We are performing the loss landscape analysis on a second dataset to characterise whether changes in their smoothness may correlate to the need for higher exploration or exploitation.
> W2: We have expanded the discussion section to reflect this as well, and are working on confirming it through further loss landscape analysis. Our hypothesis is that some loss landscapes make crossover-based approaches unable to escape local minima due to increased exploitation and feature reuse.
> W3:The variability in the performance of the different methods across loss landscapes further proves the need for specific algorithms with different exploration/exploitation tradeoffs, enabled by the efficient crossover operation presented in this work. We have expanded the discussion section accordingly, and added Appendix F, where we propose the implementation of different search strategies to encourage further research in this direction. Moreover, we are working on increasing the number of datasets and seeds to provide more statistically sound results, increasing the intra-dataset statistical significance and further showcasing the inter-dataset variability.
> Q1: The performance of STX may be explained by the random reuse of functional blocks given a diverse and performant enough initial population. Once again, we are working on increasing the number of datasets and seeds to provide more statistically sound results to ensure that the observed behaviours are not too biased by the inherent randomness of the evolutionary method employed. Note that, if block reuse were to be indeed identified as one of the most performant strategies, it could be mimicked by the (R)CSWX, as the alignment provided by these tools can be used to identify these blocks and, in theory, devise search strategies that perform this block reuse in a more structured manner.
>
> Please let us know if you have any further questions or concerns. We are eager to further improve our contribution. We would also appreciate a reconsideration of the score in the light of the changes made to the paper.

---

### Author Response · Authors · 2025-11-21
**Changes performed to the paper in response to the reviewers' feedback**

First and foremost, we want to thank you all for the thorough reviews and valuable feedback. We really appreciate the positive comments regarding the novelty and efficiency of our method, as well as the theoretical soundness of our adaptation of the Smith–Waterman alignment through the addition of constrains and recursive computing to grammar-based NAS. We are pleased to read that you considered our analyses of runtime, metric properties, and permutation invariance varied enough to provide with a clear picture of the proposed methods. Nontheless, we take your identified weaknesses and questions very seriously. In order to integrate all the feedback and answer all questions, we have made substantial changes to the paper, which are highlighted in blue in the revised PDF for easier review. In particular:

C1: We added Appendix D where we compare the distance attained between 50 pairs of randomly sampled models by using four different scoring matrices. We show that the scoring procedure selected would have some influence on the search and subsequent analysis but, given the strong correlations observed across the computed distances, we hypothesise that the overall shape of the loss landscape would be mostly maintained.
C2: We added Appendix E where we provide alignment matrices given by the RCSWX for big models (ResNet18, ResNet32, MLP_Mixer d8, MLP_Mixer d12) along with distance scores, compute times and some insights on what’s happening in real world applications.
C3: We are working on extending our analysis to 7 datasets and 5 seeds, and adding hypothesis testing to ensure that our results are statistically significant. However, we want to highlight that the focal point of this work is not to prove the performance of any given search strategy but rather that of the crossover operation, which is measured by the quality of the interpolation (which is the same as for SEPX) and the computation requirement to calculate it (which is drastically reduced). We have extended the discussion section, delving deeper into the consequences of the crossover being block-based (STX), permutation sensitive (CSWX) or permutation invariant (RCSWX) and how that translates to convergence speed and results. Moreover, we are performing our loss landscape analysis (Figure 4) on a second dataset to characterise whether the variance across the different methods’ performances correlates to changes in the loss landscape smoothness.
C4: The performance of the evolutionary method does vary greatly across datasets. This issue, which is inherent to the search strategies and not dependent on the method employed to perform the crossover itself, further proves the need for specific algorithms to properly explore the different spaces. Our goal throughout this work is to provide a fast and reliable way of performing crossover and measuring distance between models (and thus defining a space in which to interpolate) which then opens up the possibility of working with a myriad of different search strategies. We have added Appendix F where we employ the (R)CSWX to define four search algorithms based on architectural distance, multi-parent crossover, etc, exemplifying what can be achieved through the crossover operator provided in this work. Moreover, we have explored the theoretical application of the (R)CSWX to more complex grammars and graphs, showing a very wide applicability. Unfortunately, comprehensive testing of the performance of these search strategies fall outside our scope—and compute budget!

Regarding the specific feedback and questions from each reviewer, please see the individual responses.

---

### Author Response · Authors · 2025-12-03
**Final changes performed to the document**

As promised, we have extended our experiments to 5 seeds and 7 datasets to ensure that the results shown herein are of statistical significance. We have added Appendix G where we conduct significance testing to prove that, despite the very differing behavior on across datasets, none of the search strategies employed is better than the rest on average. Note that, across all experiments, we have trained and evaluated over 140.000 architectures!

We have also extended the loss landscape analysis to a second dataset, which supports our claim that each dataset presents an unique architectural loss landscape, and thus proves that chosing an appropriate search strategy with an adequate exploration-exploitation balance is crucial to converge quickly towards good models.

The main body of the paper has been updated accordingly, and further refined to accomodate all reviewers' feedback. As stated in the previous Official Comment, Appendixes D, E and F have been added precisely to answer the reviewers' main concerns, as well as an incentive for the Deep Learning community to use the tools presented herein and keep exploring the myriad of NAS algorithms and search spaces available.

We hope you find this work as novel and high-performing as the reviewers, and we would like to thank the latter once again for their thorough reviews, as we think they have improved our contribution considerably. We hope that the changes performed over the last month are deemed sufficient to reconsider our grade, and thank all chairs for their time and hard work organising such an important event.

---

### Meta-Review · Area_Chair_Atbs · 2026-01-07

**Summary:**

Based on the reviewers' feedback, the decision is to reject this submission. While the adaptation of the Smith-Waterman algorithm for Neural Architecture Search (NAS) demonstrates algorithmic ingenuity and computational efficiency, the empirical facts do not justify the engineering complexity. Reviewers z5eP, 4JE4, and hK6c consistently pointed out that the proposed method (RCSWX) frequently fails to outperform—and sometimes underperforms—simple baselines like Subtree Crossover (STX) or even mutation-only search ("No Crossover"). For this conference, we encourage reducing engineering overhead unless it yields very significant results; here, the authors’ own significance testing in the rebuttal confirmed that their method provides no statistically significant advantage over standard baselines.

**Reviewer Concerns:**

Addressed:

    ◦ Scalability and Speed: Reviewers 4JE4 and hK6c appreciated the runtime analysis demonstrating that (R)CSWX scales significantly better than Graph Edit Distance (SEPX) for large architectures.
    ◦ Statistical Rigor: In response to JJpP and hK6c, the authors expanded experiments from 3 to 5 seeds and increased the number of datasets to 7.
    ◦ Parameter Sensitivity: Concerns from hK6c regarding the "skewness" parameter and scoring matrices were addressed via new sensitivity analyses in Appendix D.

Outstanding:

    ◦ Performance relative to Baselines: This is the critical outstanding "fact" check. z5eP, 4JE4, and hK6c all noted that the complex RCSWX operator often performs worse than the simpler CSWX or the "No Crossover" baseline. The authors' rebuttal confirmed that there is no statistical difference between the proposed method and the baselines (p=0.0997), failing to validate the core motivation of the paper.
    ◦ Comparison to State-of-the-Art (SoTA): JJpP highlighted the absence of comparisons with current SoTA NAS approaches. The authors argued their method is a "tool" rather than a competitor for SoTA, but this limits the paper's impact and practical relevance for the conference.
    ◦ Motivation for Complexity: hK6c noted that since performance evaluation (training) is the bottleneck in NAS, not the crossover operation, a faster crossover that yields lower-quality architectures (as RCSWX does on AddNIST) is not a justifiable trade-off.

**Reviewer Scores:**

• Reviewer z5eP (Score: 4): Unlikely to change. The reviewer’s primary concern was that performance gains were insignificant and unstable across datasets, a fact confirmed by the authors' expanded analysis in Appendix G.

• Reviewer 4JE4 (Score: 4): Unlikely to change. The reviewer noted that RCSWX offering "no empirical improvement on average" was a major weakness. The rebuttal provided data that reinforced this observation rather than refuting it.

• Reviewer hK6c (Score: 4): Unlikely to change. The reviewer fundamentally questioned the utility of a "faster" search metric that results in lower performance. The authors' pivot to framing the method as an "analysis tool" is unlikely to warrant a higher score for a search algorithm paper.

• Reviewer JJpP (Score: 2): Unlikely to change. This reviewer explicitly maintained their rating after the rebuttal, citing the lack of rigorous theoretical analysis for the new appendices and the continued absence of SoTA comparisons.

---

### Decision · Program_Chairs · 2026-01-26

Reject